# DynPoint: Dynamic Neural Point For View Synthesis

**Kaichen Zhou, Jia-Xing Zhong, Sangyun Shin, Kai Lu,**
**Yiyuan Yang**, **Andrew Markham**, **Niki Trigoni**
Department of Computer Science
University of Oxford
{rui.zhou, jiaxing.zhong, sangyun.shin, kai.lu}@cs.ox.ac.uk
{yiyuan.yang, andrew.markham, niki.trigoni}@cs.ox.ac.uk

## Abstract

The introduction of neural radiance fields has greatly improved the effectiveness of view synthesis for monocular videos. However, existing algorithms face difficulties when dealing with uncontrolled or lengthy scenarios, and require extensive training time specific to each new scenario. To tackle these limitations, we propose DynPoint, an algorithm designed to facilitate the rapid synthesis of novel views for unconstrained monocular videos. Rather than encoding the entirety of the scenario information into a latent representation, DynPoint concentrates on predicting the explicit 3D correspondence between neighboring frames to realize information aggregation. Specifically, this correspondence prediction is achieved through the estimation of consistent depth and scene flow information across frames. Subsequently, the acquired correspondence is utilized to aggregate information from multiple reference frames to a target frame, by constructing hierarchical neural point clouds. The resulting framework enables swift and accurate view synthesis for desired views of target frames. The experimental results obtained demonstrate the considerable acceleration of training time achieved - typically an order of magnitude - by our proposed method while yielding comparable outcomes compared to prior approaches. Furthermore, our method exhibits strong robustness in handling long-duration videos without learning a canonical representation of video content.

## 1 Introduction

The computer vision community has directed significant attention towards novel view synthesis (VS) due to its potential for both emerging techniques in artificial reality and also to enhance a machine's ability to comprehend the appearance and geometric properties of target scenarios [37, 8, 39]. State-of-the-art techniques leveraging neural rendering algorithms, as demonstrated in studies such as [37, 58, 61], have successfully attained photorealistic reconstruction of views in static scenarios. However, the dynamic characteristics inherent to most real-world scenarios present a formidable challenge to the suitability of existing approaches that rely on the epipolar geometric relationship, traditionally applicable to static scenarios [22, 45].

Recent studies have primarily focused on the synthesis of views in dynamic scenarios by employing one or multiple multilayer perceptrons (MLPs) to encode the essential spatiotemporal information of the scene [45, 31, 43, 42]. In one approach, a latent representation is generated to encompass the comprehensive per-frame details of the target video [31, 16, 62]. Although this method is capable of producing visually realistic results, its applicability is limited to short videos due to the constrained memory capacity of MLPs or other representation mechanisms [32]. Alternatively, another approach seeks to construct a latent representation of canonical scenarios and establish correspondences between individual frames and the canonical scenario [45, 43, 63]. This alternative approach allows for the processing of long-term videos; however, it imposes specific requirements on the video

37th Conference on Neural Information Processing Systems (NeurIPS 2023).

characteristics being learned. For instance, the videos should consistently feature similar objects across frames, and some algorithms may require prior knowledge about the video [60, 44].

In order to address this challenge, we introduce DynPoint, a novel approach designed to achieve efficient view synthesis of lengthy monocular videos without the need for learning a latent canonical representation. Unlike conventional methods that encode information implicitly, DynPoint employs an explicit estimation of consistent depth and scene flow for surface points. These estimates are subsequently utilized to aggregate information from reference frames into the target frame. Subsequently, hierarchical neural point clouds are constructed based on the aggregated information. This hierarchical point cloud set is then employed to synthesize views of the target frame. Our contributions can be summarized as follows:

- We introduce a novel module to estimate consistent depth information for monocular video with the help of a proposed regularization and training strategy.
- We propose an efficient approach to estimate smooth scene flow between adjacent frames with the proposed training strategy by leveraging the estimated consistent depth maps.
- We present a representation to aggregate information from reference frames to target frame, facilitating rapid view synthesis of the target frame within a monocular video.
- Comprehensive experiments are conducted on datasets including Nerfie, Nvidia, HyperNeRF, Iphone, and Davis showcasing the speed and accuracy of DynPoint for view synthesis.

## 2 Related Works

### 2.1 Static View Synthesis

The generation of photo-realistic views from arbitrary input viewing angles has been a longstanding challenge in the field of computer vision. In the scenario of static scenes, early approaches addressed the challenge of synthesizing photo-realistic views by employing local warping techniques to handle densely sampled views [19, 30, 2, 5, 7]. Additionally, the gradient domain was utilized to handle the single-view case [29]. In order to tackle the challenges associated with view synthesis, several subsequent works have been developed. These works aim to address issues such as reflections, repetitive patterns, and untextured regions [28, 15, 24, 14, 48, 9, 34, 36, 41, 50, 51, 52, 53, 59, 69]. In more recent developments, researchers have explored the representation of scenes as continuous neural radiance fields (NeRF) using fully connected neural networks. These works, such as [38, 1, 66, 25], have demonstrated remarkable outcomes with a trade-off between accuracy and computational complexity (i.e., one network for one scene). To tackle the computational burden associated with neural radiance fields, recent works have focused on developing approaches that generalize the representation across multiple scenes using a single network. These methods employ various techniques, such as fully-convolutional architectures [65], plane-swept cost volumes [8], image-based rendering [58], disentanglement of shape and texture [27], utilization of local features in 2D [56], as well as generative methods [6, 49, 40, 20, 3].

### 2.2 Dynamic View Synthesis

The emergence of static scene advancements has spurred interest in the exploration of view synthesis for dynamic scenes within the field. Previous research efforts have expanded upon studies conducted in static scenes, incorporating elements such as globally coherent depth [64] and 3D mask volume [33]. Recent research direction builds upon the concept of NeRF and extends it to dynamic scenes by integrating time into the learning process of spatio-temporal radiance fields. One category of research focuses on the assumption of a canonical scenario that spans the entire video [4, 10, 21]. Deformable radiance field-based approaches, as described in previous works [45, 55, 42, 13, 35], employ temporal warping techniques to adapt a base NeRF representation of the canonical space for dynamic scenes. This enables the synthesis of novel views in the context of long monocular videos. Another category of algorithms aims to encode the temporal dynamics of the scene into a global representation. [31] employs a MLP to model the 3D dense motion, resulting in a spatial-temporal NeRF. Building upon this, [12, 57, 16] demonstrate that introducing additional regularization techniques that encourage consistency and physically plausible solutions can enhance the accuracy of view reconstruction. The first category of methods, while achieving impressive photorealistic results, is constrained by the requirement for object-centric videos, thus limiting their generalizability to diverse scenarios.

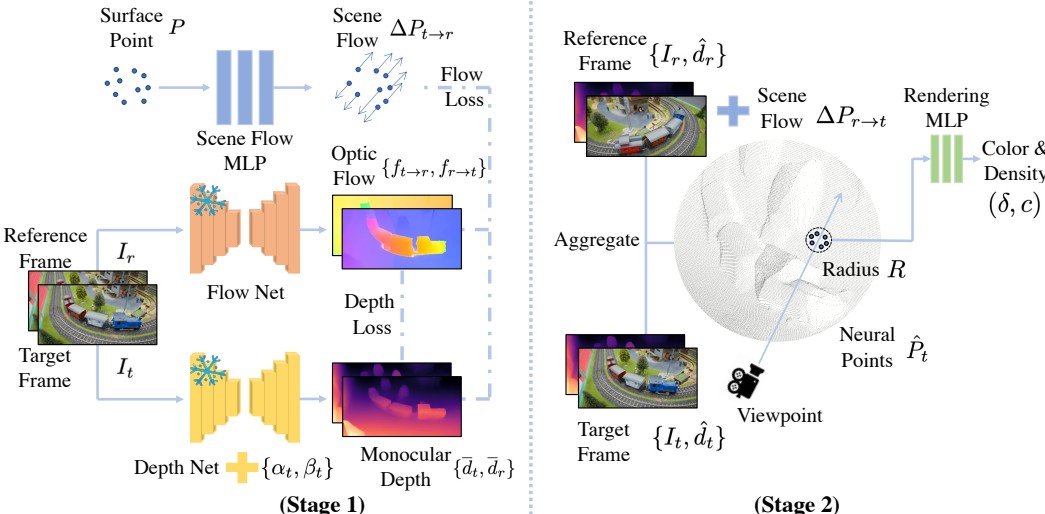

Figure 1: **Structure of DynPoint.** The **Stage 1** shows the pipeline of consistent depth estimation in Sec. 3.2 and scene flow estimation in Sec. 3.3. Initially, the frames are employed in the Flow Net, Depth Net, and Scale Parameters to produce optic flows and depth. Then, surface points are calculated based on the estimated depth and utilized in the Scene Flow MLP. The **Stage 2** shows the process of information aggregation presented in Sec. 3.4. Neural Point Clouds is firstly generated based on pre-computed scene flow. The Rendering MLP utilizes all neural points located within a specified radius from the queried point as inputs to predict the final color and density.

Conversely, the second category of methods encounters difficulties when dealing with long videos, leading to limitations in effectively handling such scenarios. To address the aforementioned issue, our algorithm focuses on achieving novel synthesis by aggregating information from the reference frame to the target frame. This information aggregation process is accomplished by explicitly modeling the object movement and depth information. As a result, our algorithm exhibits significant advancements in both accuracy and speed.

## 3 Methodology

### 3.1 Overview

Our algorithm is designed to realize view synthesis for a dynamic scenario by utilizing a monocular video $\{I_1, I_2, ..., I_T\}$. The frames in the video, denoted by $I_t$, are captured by a known camera $\mathbf{C}_{t,c} = \{\mathbf{K}_{t,c} | [\mathbf{R}_{t,c}, \mathbf{t}_{t,c}]\}$, where $c$ denotes known camera viewpoint. The objective is to generate the novel view from a specified viewpoint $q$ at a desired time frame $\mathbf{C}_{t,q} = \{\mathbf{K}_{t,q} | [\mathbf{R}_{t,q}, \mathbf{t}_{t,q}]\}$.

Consistent with previous researches [31, 16, 17, 32, 35, 62], we adopt a training paradigm in which our model is trained on the input monocular video with the assistance of pre-trained optic flow and monocular depth models [47, 54, 26]. During the training process, the RGB information obtained from the observed viewpoint is utilized as the supervision signal, without relying on any canonical information. Subsequently, the trained model is evaluated on the task of synthesizing corresponding RGB, depth and scene flow information for unobserved viewpoints.

In contrast to previous methods that utilize one or multiple MLPs to encode the 3D information of each frame, our work focuses on establishing correspondence relationships, i.e., scene flow, between the 3D surface points of the current frame and those of adjacent frames. We can infer 3D information about unobserved points of the current frame by aggregating the information from adjacent frames.

To realize the aforementioned concept, our proposed model should undertake three key tasks. The first task is to estimate depth information consistently for each frame, as outlined in Sec. 3.2. The second task involves learning the 3D scene flow between the current frame and its adjacent frames, as detailed in Sec. 3.3. These two processes are demonstrated in the stage 1 of Fig. 1. The final task is

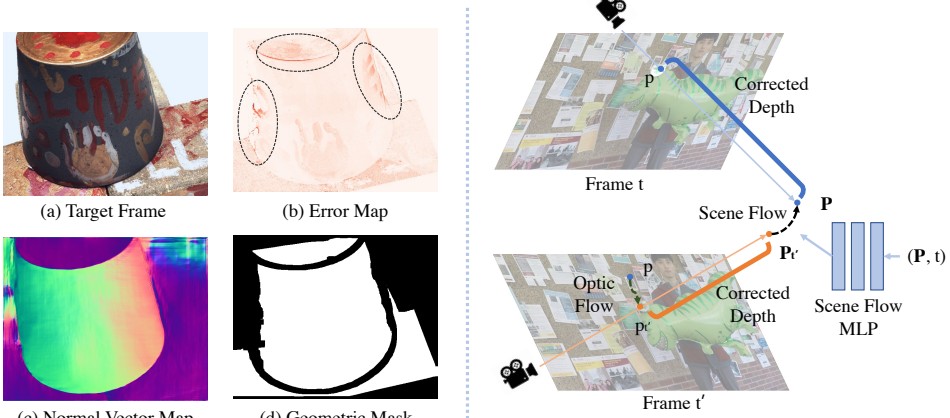

(a) Target Frame        (b) Error Map

(c) Normal Vector Map      (d) Geometric Mask

Figure 2: **Demonstration of Geometric Edge Mask and Scene Flow Estimation.** The **left** section depicts the conceptual basis for designing the Geometric Edge Mask. The **right** part demonstrates the construction of the scene flow objective function shown in Sec. 3.3.

to aggregate information based on learned correspondence and subsequently use this information to realize view synthesis, as discussed in Sec. 3.4. This process is shown in the stage 2 of Fig. 1.

## 3.2 Consistent Depth Estimation

Despite the ability of current monocular depth methods [47, 46] to produce accurate depth priors, the predicted depth maps $d'_t$ suffer from scale-variance and shift problems when compared to the ground truth depth [18, 68]. This characteristic renders $d'_t$ inconsistent across the temporal axis of monocular video. Moreover, since the optic flow $f_{t \to t'}$ between frames could be predicted from a pretrained model, a depth value $\bar{d}_t$ can be computed using the triangulation relation between frames, along with the camera parameters and Mid-point method. Compared to $d'_t$, $\bar{d}_t$ can have consistent scaling across frames. However, $\bar{d}_t$ heavily relies on $f_{t \to t'}$ and Epipolar constraint, and it can only generate accurate depth information for a limited portion of the frame. The first module of DynPoint aims to combine $d'_t$ and $\bar{d}_t$ to generate the final consistent depth estimation $\hat{d}_t$.

**Regularization:** To address this issue, DynPoint identifies the accurate region of $\bar{d}_t$ by utilizing three masks: the corresponding mask, the geometric edge mask, and the dynamic object mask.

Correspondence Mask $\mathcal{M}_{c,t \to t'}$: The purpose of the optic flow is to capture the pixel-wise correspondence between two frames, making it accurate only for the corresponding regions. These regions can be identified by masking out areas of occlusion caused by both ego-motion and object movement. This Correspondence Mask could be formulated as follow:

$$\mathcal{M}_{c,t \to t'} = \begin{cases} 0 & if \quad |f_{t \to t'}(p) + f_{t' \to t}(p + f_{t \to t'}(p))| > \epsilon_c, \\ 1 & if \quad |f_{t \to t'}(p) + f_{t' \to t}(p + f_{t \to t'}(p))| \leq \epsilon_c, \end{cases} \tag{1}$$

where $\epsilon_c$ is a predefined threshold for the correspondence mask.

Geometric Edge Mask $\mathcal{M}_g$: Furthermore, it has been observed that the reliability of $\bar{d}_t$ diminishes when applied to geometric edges as in Fig. 2, particularly in areas where the optic flow exhibits non-smooth characteristics. To compute the mask $\mathcal{M}_g$, we first estimate the normal vector map of the surface $n = (-\frac{dz}{dx}, -\frac{dz}{dy}, 1)/||(-\frac{dz}{dx}, -\frac{dz}{dy}, 1)||$ with the help of $\bar{d}_t$. Then we apply the Canny edge detector [11] on the $n \in \mathbb{R}^{H \times W}$ to generate the estimation of geometric edge. The intuition of the geometric edge mask is demonstrated in the left part of Fig. 2. From the error map shown in Fig. 2(b), it is evident that the errors primarily manifest around the geometric boundaries of the scenario.

Dynamic Object Mask $\mathcal{M}_d$: In addition to the two masks mentioned above, a Dynamic Object Mask is necessary to exclude the dynamic regions of the frame where the triangulation relationship does not hold. Similar to [16], we combine the MASK R-CNN [23] with the Sampson error to generate the $\mathcal{M}_d$. To obtain a valid mask with high accuracy, we apply *image erosion* on the inverse of $\mathcal{M}_d$ and

$\mathcal{M}_c$ to mitigate inaccurate boundary detection. We also perform *image dilation* on $\mathcal{M}_g$ to include the surrounding region of the geometric edge.

**Objective Function:** Based on the above three masks, the reliable region of the $\overline{d}_t$ could be masked out by using the final mask $\mathcal{M}_f = \mathcal{M}_c \cap \mathcal{M}_g \cap \mathcal{M}_d$. To combine the information of $d'_t$ and $\overline{d}_t$, we assume that the consistent depth map $\hat{d}_t \in \mathbb{R}^{H \times W}$ could be approximated by using $d'_t$, a scale variable $\alpha_t \in \mathbb{R}$ and a shift variable $\beta_t \in \mathbb{R}$. The parameters $\alpha_t$ and $\beta_t$ can be generated by optimizing:

$$\alpha_t, \beta_t = \text{argmin}\, \mathcal{M}_f \odot |\overline{d}_t - (\alpha_t d'_t + \beta_t)|. \tag{2}$$

**Training Strategy:** Dependent solely on the nearest frame for the computation of $\overline{d}_t$ would lack sufficient reliability due to the combined influence of the accuracy of the camera matrix and the accuracy of the optic flow on the triangulation process. We employ a series of adjacent $2K$ frames to calculate the triangulated depth set denoted as $\{\overline{d}_t^k\}_{k=1}^{2K}$. Instead of directly utilizing all $\{\overline{d}_t^k\}_{k=1}^{2K}$ in Eqn. 2, we perform a reevaluation by recomputing the intersection mask as $\mathcal{M}_f = \mathcal{M}_f^1 \cap ... \cap \mathcal{M}_f^{2K}$, which further refines the triangulated depth $\overline{d}_t$, ensuring the appropriate scale constraint for $d'_t$. It's worth noting that pose estimation may not be accurate for dynamic scenarios in certain datasets, as it relies on COLMAP-based estimation. In such cases, we utilize the algorithm presented in our Supplementary material to enhance and fine-tune the pose estimation.

### 3.3 Scene Flow Estimation

Combined with estimated consistent depth map $\hat{d}_t$ and optic flow $f_{t \to t'}$, DynPoint also aims to infer the scene flow $s_{t \to t'}$ to build the 3D correspondence between the current frame and adjacent frames. Unlike previous works [42, 45, 55, 13], which estimate the trajectory of all points (hundreds of sampled points on the ray of each point) in the scenario, DynPoint only infers the trajectory of surface point (one point on the ray of each point) of the frame to accelerate both training and inference process. To realize this process, we use a MLP to estimate the scene flow, which can be written as $\Delta P_{t \to t+1}, \Delta P_{t \to t-1} = F_\theta(P, t)$ where $P \in \mathcal{R}^3$ denotes input 3D point; $\Delta P_{t \to t'}$ denotes the trajectory of $P$ from $t$ to $t'$. The weight $\theta$ can be optimized by using the relationship among the depth $\hat{d}_t$, the optic flow $f_{t \to t'}$ and the scene flow $s_{t \to t'}$.

**Objective Function:** Given a pixel $p_t$ in frame $t$, its corresponding pixel in frame $t'$, denoted as $p_{t'}$, can be obtained by adding the 2D flow $f_{t \to t'}(p_t)$ to $p_t$. Additionally, utilizing the depth map $\hat{d}_t$ and camera matrix $\mathbf{C}_t$, which are available at frame $t$, the 3D point corresponding to $p_t$ can be expressed as $P_t = \mathbf{R}_t \mathbf{K}_t^{-1} \hat{d}_t(p_t) p_t + \mathbf{t}_t$. The same method can be used to compute $P_{t'}$. Thus, we've:

$$s_{t \to t'}(p_t) = \hat{d}_{t'}(p_t + f_{t \to t'}(p_t)) \mathbf{R}_{t'} \mathbf{K}_{t'}^{-1}(p_t + f_{t \to t'}(p_t)) + \mathbf{t}_{t'} - P_t. \tag{3}$$

This process is demonstrated in the right part of Fig. 2. For static part masked by $\mathcal{M}_d$, we set $s_t(p_t) = 0$. For the dynamic part, the loss function can be written as

$$\mathcal{L}_s = \sum_{p_t \in \mathcal{M}_c \cap \mathcal{M}_g \cap \neg \mathcal{M}_d} |s_{t \to t'}(p_t) - \Delta P_{t \to t'}|. \tag{4}$$

In order to enhance the accuracy of scene flow estimation for the dynamic elements within the scenario, we employ the cycle constraint, a well-established technique utilized in prior studies [16, 35]. The cycle constraint can be expressed as follows:

$$\mathcal{L}_c = \sum_{p_t \in \neg \mathcal{M}_d} |\Delta P_{t \to t+k} + \Delta P_{t+k \to t}(P_t + \Delta P_{t \to t+k})|. \tag{5}$$

**Training Strategy:** We have observed that optimizing Eqn. 4 solely with the near frames, where $t' = t - 1$ or $t + 1$, does not produce accurate results for further information aggregation. To reconstruct the 3D information of the current frame, it is necessary to compute the correspondence between the current frame and $2K$ adjacent frames, in order to aggregate sufficient information. During the training process, we compute the scene flow between frame $t$ and its $2K$ adjacent frames, where $K \in \{1, ..., K\}$. We then utilize Eqn. 4 to form the loss function between current frame and $2K$ adjacent frames. The scene flow $\Delta P_{t \to t+k}$ could be written as:

$$\Delta P_{t \to t+k} = F_\theta(P_t, t)[0] + ... + F_\theta(P_t + \Delta P_{t \to t+k-1}, t + k - 1)[0]. \tag{6}$$

Table 1: **Novel View Synthesis Results on Nvidia Dataset.** We report the average PSNR and LPIPS results with comparisons to existing methods on Nvidia dataset [64]. * denotes the number adopted from DynamicNeRF [16]. The best performance is **highlighted**. The second-best is also emphasized.

| PSNR ↑ / LPIPS ↓ | Jumping | Skating | Truck | Umbrella | Balloon1 | Balloon2 | Playground | Average |
|---|---|---|---|---|---|---|---|---|
| NeRF* [38] (∼ **24 hours**) | 20.99 / 0.305 | 23.67 / 0.311 | 22.73 / 0.229 | 21.29 / 0.440 | 19.82 / 0.205 | 24.37 / 0.098 | 21.07 / 0.165 | 21.99 / 0.250 |
| NeRF + time*[38](∼ **24 hours**) | 18.04 / 0.455 | 20.32 / 0.512 | 18.33 / 0.382 | 17.69 / 0.728 | 18.54 / 0.275 | 20.69 / 0.216 | 14.68 / 0.421 | 18.33 / 0.427 |
| D-NeRF [45] (> **20 hours**) | 22.36 / 0.193 | 22.48 / 0.323 | 24.10 / 0.145 | 21.47 / 0.264 | 19.06 / 0.259 | 20.76 / 0.277 | 20.18 / 0.164 | 21.48 / 0.232 |
| NSFF* [31](∼ **24 hours**) | 24.65 / 0.151 | 29.29 / 0.129 | 25.96 / 0.167 | 22.97 / 0.295 | 21.96 / 0.215 | 24.27 / 0.222 | 21.22 / 0.212 | 24.33 / 0.199 |
| DynamicNeRF* [16] (> **36 hours**) | 24.68 / 0.090 | **32.66 / 0.035** | 28.56 / 0.082 | 23.26 / 0.137 | 22.36 / 0.104 | 27.06 / **0.049** | 24.15 / 0.080 | 26.10 / 0.082 |
| HyperNeRF [43] (> **24 hours**) | 18.34 / 0.302 | 21.97 / 0.183 | 20.61 / 0.205 | 18.59 / 0.443 | 13.96 / 0.530 | 16.57 / 0.411 | 13.17 / 0.495 | 17.60 / 0.367 |
| TiNeuVox [13] (∼ 45 mins) | 20.81 / 0.247 | 23.32 / 0.152 | 23.86 / 0.173 | 20.00 / 0.355 | 17.30 / 0.353 | 19.06 / 0.279 | 13.84 / 0.437 | 19.74 / 0.285 |
| RoDYN [35] (> **36 hours**) | **25.66 / 0.071** | 28.68 / 0.040 | 29.13 / 0.063 | 24.26 / 0.089 | 22.37 / 0.103 | 26.19 / 0.054 | 24.96 / 0.048 | 25.89 / **0.065** |
| DynPoint (∼ **30 mins**) | 24.69 / 0.097 | 31.34 / 0.045 | **29.30 / 0.061** | **24.59 / 0.086** | **22.77 / 0.099** | **27.63 / 0.049** | **25.37 / 0.039** | **26.53** / 0.068 |

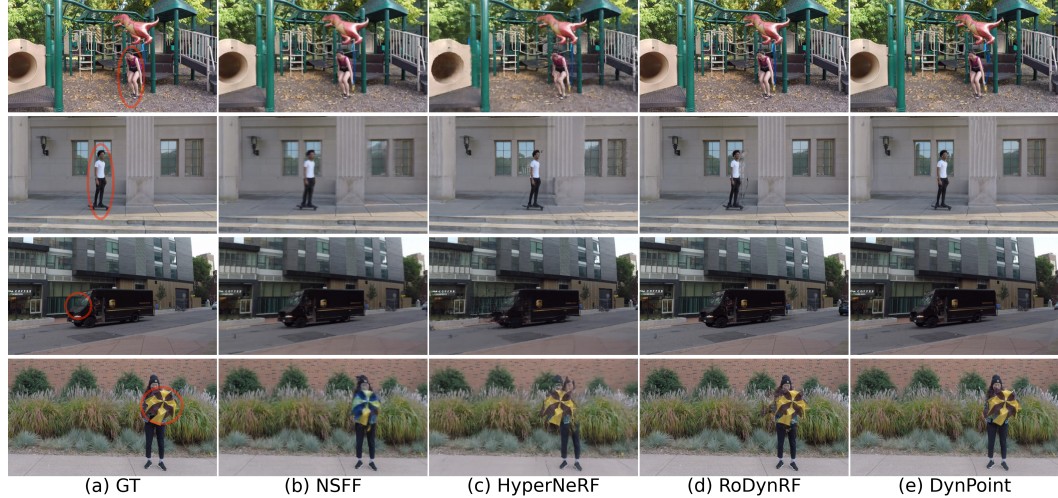

|  (a) GT  |  (b) NSFF  |  (c) HyperNeRF  |  (d) RoDynRF  |  (e) DynPoint  |

Figure 3: **Demonstration of View Synthesis Results on Nvidia Dataset.** This demonstration compares the view synthesis outcomes of DynPoint with those of NSFF, HyperNeRF, and RoDynRF.

It's important to mention that achieving better results can be accomplished by fine-tuning the refinement layers of the pre-trained depth network.

### 3.4  View Synthesis

**Information Aggregation**: In order to collate data from the $2K$ reference frames to the target frame $t$, the pixels in the reference frame $p_{t+k}$ are transformed into 3D space as $P_{t+k}$ through the utilization of the corresponding depth value $\hat{d}_{t+k}$ and the matrix $\mathbf{C}_{t+k}$. Next, the DynPoint evaluates the scene flow $\Delta P_{t+k\rightarrow t}$ between frame $t$ and frame $t+k$ by implementing Eqn. 6. By utilizing both the scene flow $\Delta P_{t+k\rightarrow t}$ and the 3D point $P_{t+k}$, we are able to calculate the 3D point $P_{t+k\rightarrow t}$ that corresponds to reference frame $t+k$ at the target frame $t$. This is achieved by applying the following computation: $P_{t+k\rightarrow t} = P_{t+k} + \Delta P_{t+k\rightarrow t}$. Additionally, a pre-trained 2D convolutional neural network (CNN) is utilized to generate a 2D image feature vector $\mathbf{f}$, which is subsequently allocated to each of the points corresponding to pixels.

Finally, we propose the introduction of a hierarchical neural point cloud construction approach to enhance the "perception field" of individual points. This technique aims to address the problem of surface irregularities that may arise from using current monocular depth estimation methods in multi-view depth map fusion. Thus, the point cloud of current frame $t$ could be generated by combining $2K + 1$ point clouds as follows:

$$\hat{P}_t = \{ \sum_{k=-K}^{K} P_{t+k\rightarrow t}^h, P_t^h, \sum_{k=-K}^{K} \mathbf{f}(P_{t+k\rightarrow t}^h), \mathbf{f}(P_t^h)\}_{h=1}^H, \tag{7}$$

where $H$ is the number of hierarchical levels. In our case, we set $H = 3$. At each step within this framework, we perform a downsampling operation on the neural point cloud (the downsampling procedure is implemented within the point cloud generation process, whereby the depth map and scene map are downsampled using linear interpolation technique; it is important to note that when

Table 2: **Novel View Synthesis Results on Nerfie Dataset.** We report the average PSNR and LPIPS results with comparisons to existing methods on Nerfie dataset [42].

| PSNR ↑ / LPIPS ↓ | CURLS | TOBY SIT | TAIL | BROOM | Average |
|---|---|---|---|---|---|
| NeRF [38] | 14.40 / 0.616 | 22.80 / 0.463 | 23.00 / 0.571 | 21.00 / 0.667 | 20.30 / 0.579 |
| NeRF + time[38] | 17.30 / 0.539 | 19.40 / 0.385 | 24.90 / 0.404 | 21.90 / 0.576 | 20.87 / 0.476 |
| NSFF [31] | 18.00 / 0.432 | **26.90** / 0.208 | **30.60** / 0.245 | **28.20** / **0.202** | 25.93 / 0.272 |
| Nerfies [42] | **24.90** / **0.312** | 22.80 / **0.174** | 23.60 / **0.175** | 21.00 / 0.270 | 23.08 / **0.233** |
| DynPoint | 24.33 / 0.339 | 24.90 / 0.186 | 29.12 / 0.218 | 27.28 / 0.222 | **26.41** / 0.241 |

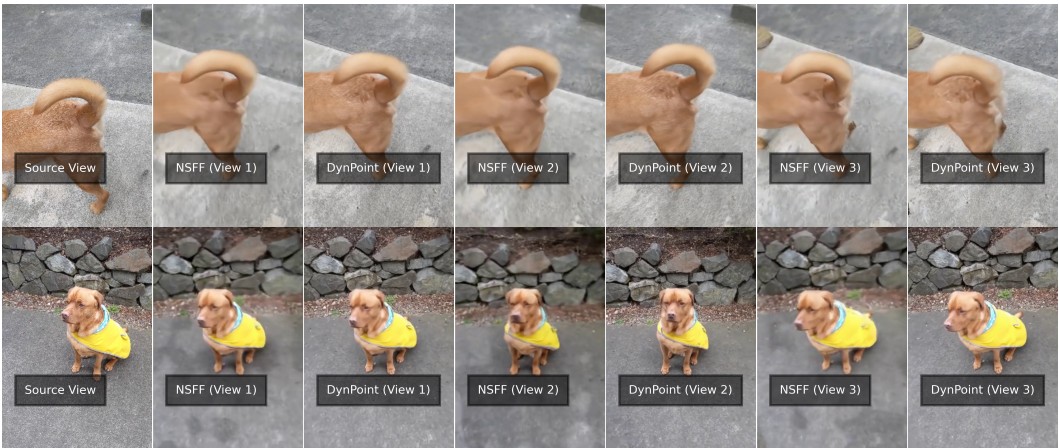

Figure 4: **Demonstration of View Synthesis Results on Nerfie Dataset.** This demonstration compares the view synthesis outcomes of DynPoint with those of NSFF.

downsampling the depth map, the intrinsic matrix should also be adjusted accordingly to maintain accurate spatial information.)

**View Synthesis**: Given a 3D position $q$ and view direction $d$, we leverage $M$ proximate neural points within a radius of $R$ to generate the corresponding density and color data $\delta, c$ as in [61]. This process is shown in the right part of Fig. 1. This can be expressed as follows:

$$(\delta, c) = F_\phi(q, d, \hat{P}_t^1, \mathbf{f}_t^1, \gamma_t^1, ..., \hat{P}_t^M, \mathbf{f}_t^M, \gamma_t^M). \tag{8}$$

where $\gamma_t^1$ is the per-point confidence introduced in [61]. Finally, we make use of the rendering process in [38] to predict final RGB value $C$ and depth $D$ as:

$$C = \sum_{j=1}^{N} \tau_j (1 - exp(-\sigma_j \delta_j)) c_j, \tag{9}$$

where $\tau_j = exp(-\sum_{t=1}^{j-1} \sigma_t \delta_t)$; $\delta_j$ denotes the distance between adjacent shading samples; $c_j$ is the color information and $\delta_j$ is the density information. The L2 loss function is used to supervise our rendered pixel values similar to the setting of [37]. For further information regarding the network architecture, please consult our supplementary materials.

### 3.5 Discussion

The purpose of this section is to explicate the dissimilarities between the DynPoint and a recently published concurrent work, namely DynIBaR [32] which was released in March 2023 and currently lacks available code. Both DynIBaR and DynPoint harness information aggregation mechanisms to realize the synthesis of novel views. However, DynIBaR predominantly centers around the aggregation of information through two-dimensional (2D) pixel units. This approach draws inspiration from image-based rendering principles, entailing the synthesis of novel perspectives from a collection of reference images via a weighted fusion of reference pixels. In contrast, DynPoint's focal point lies in the information aggregation achieved by constructing three-dimensional (3D) neural point clouds. The final novel view synthesis is realized by using neural points surrounding queries' position.

Table 3: **Novel View Synthesis Results of HyperNeRF Dataset.** We report the average PSNR and LPIPS results with comparisons to existing methods on HyperNeRF dataset [43].

| PSNR ↑ / LPIPS ↓ | Broom 197 Frames | 3D printer 207 Frames | Chicken 164 Frames | Expressions 259 Frames | Peel Banana 513 Frames | Average |
|---|---|---|---|---|---|---|
| NSFF [31] | 26.10 / 0.284 | 27.70 / 0.125 | 26.90 / 0.106 | 26.70 / 0.157 | 24.60 / 0.198 | 26.40 / 0.174 |
| Nerfies [42] | 19.20 / 0.325 | 20.60 / 0.108 | 26.70 / 0.078 | 21.80 / 0.150 | 22.40 / 0.147 | 22.10 / 0.162 |
| Hyper-NeRF [43] | 20.60 / 0.613 | 21.40 / 0.212 | 27.60 / 0.108 | 22.00 / 0.196 | 24.30 / 0.170 | 23.20 / 0.260 |
| DynPoint | 27.40 / 0.248 | 27.60 / 0.163 | 28.10 / 0.089 | 27.90 / 0.147 | 26.50 / 0.129 | 27.50 / 0.155 |

Table 4: **Novel View Synthesis Results of Iphone Dataset.** We compare the mPSNR and mSSIM scores with existing methods on the iPhone dataset [17].

| mPSNR ↑ / mSSIM ↑ | Apple | Block | Paper-windmill | Space-out | Spin | Teddy | Wheel | Average |
|---|---|---|---|---|---|---|---|---|
| NSFF [31] | 17.54 / 0.750 | 16.61 / 0.639 | 17.34 / 0.378 | 17.79 / 0.622 | 18.38 / 0.585 | 13.65 / 0.557 | 13.82 / 0.458 | 15.46 / 0.569 |
| Nerfies [42] | 17.64 / 0.743 | 17.54 / 0.670 | 17.38 / 0.382 | 17.93 / 0.605 | 19.20 / 0.561 | 13.97 / 0.568 | 13.99 / 0.455 | 16.45 / 0.569 |
| HyperNeRF [43] | 16.47 / 0.754 | 14.71 / 0.606 | 14.94 / 0.272 | 17.65 / 0.636 | 17.26 / 0.540 | 12.59 / 0.537 | 14.59 / 0.511 | 16.81 / 0.550 |
| DynPoint | 17.78 / 0.743 | 17.67 / 0.667 | 17.32 / 0.366 | 17.78 / 0.603 | 19.04 / 0.564 | 13.95 / 0.551 | 14.72 / 0.515 | 16.89 / 0.572 |

Moreover, DynPoint introduces an efficacious strategy for the seamless integration of monocular depth estimation within the ambit of monocular video view synthesis. In contrast to DynIBaR, which endeavors to model the trajectory of all samples (128) traversing each ray, DynPoint exclusively focuses on the trajectory of surface points, thereby yielding a substantial acceleration in both training and inference stage.

# 4 Experiment

## 4.1 Experimental Setting

To evaluate the view synthesis capabilities of DynPoint, we performed experiments on four extensively utilized datasets, namely Nvidia dataset in [31], Nerfie in [42], HyperNeRF in [43] and Iphone in [17]. It is noteworthy to mention that the official website of Nerfie [42] only provides four specific scenarios. Consequently, our experiments were solely conducted on provided four scenarios as in [17]. Additionally, we also assessed DynPoint's performance on a recent dataset Iphone [17], which specifically addresses the challenge of camera teleportation. Furthermore, we examined the efficacy of monocular depth estimation and scene estimation by visualizing the results obtained from the Davis dataset, as in [67].

We conducted a comparative analysis of our work with several recent methods, based on the reported results in their original papers or the reimplementation results of their official code. The methods we compared with include NeRF [38], NeRF + time [38] (which directly utilizes embedded time information as an input to encode all information of the target dynamic scenario), D-NeRF [45], NSFF [31], DynamicNeRF [16], HyperNeRF [43], Nerfie [42], TiBeuVox [13], and RoDYN [35]. In our research, we employed the pretrained Deep Pruning Transformer (DPT) network in [46], for monocular depth estimation. For optic flow estimation, we utilized the pretrained FlowFormer model [26]. To facilitate the fine-tuning process, we initialized the weights of the Rendering MLP by pretraining it on the DTU dataset, employing a similar training set to that used in [61].

The per-scenario training time is shown in Tab. 1. A notable observation was made regarding the reduced per-scenario training time of DynPoint in comparison to other algorithms. This enhancement can be attributed to the implementation of a two-step strategy. In the initial stage, the optimization process focuses on a limited set of parameters pertaining to monocular depth estimation and scene flow estimation. Subsequently, for the second stage, leveraging the outcomes obtained from the first stage, the generation of the neural point cloud for subsequent view synthesis is achieved successfully. Furthermore, the pretraining stage also contributes to the efficiency of our view synthesis stage, requiring only a few iterations to produce desirable results on novel scenarios.

**Nvidia** (Tab. 1): During this experiment, it was observed that DynPoint exhibited superior performance compared to other algorithms in terms of peak signal-to-noise ratio (PSNR). Additionally, DynPoint achieved the second highest ranking among all algorithms when evaluated based on the Learned Perceptual Image Patch Similarity (LPIPS) metric. Even for scenarios involving multiple objectives, such as Jumping, DynPoint demonstrates reasonable performance without requiring the learning of any canonical representation or extensive training time. The results of specific partial

Table 5: **Ablation studies of Nvidia Dataset.** We report the average PSNR and LPIPS results with comparisons to existing methods on Nvidia dataset [64].

| PSNR ↑ / LPIPS ↓ | Jumping | Skating | Truck | Umbrella | Balloon1 | Balloon2 | Playground | Average |
|---|---|---|---|---|---|---|---|---|
| w/o multiple-step strategies | 15.01 / 0.711 | 17.43 / 0.691 | 15.36 / 0.606 | 14.90 / 0.837 | 13.98 / 0.583 | 15.46 / 0.624 | 11.35 / 0.613 | 14.78 / 0.666 |
| w/ K = 3 | 20.22 / 0.262 | 22.53 / 0.274 | 22.38 / 0.291 | 19.13 / 0.383 | 16.59 / 0.413 | 16.92 / 0.459 | 14.83 / 0.416 | 18.94 / 0.357 |
| w/o Hierarchical Point Cloud | 23.82 / 0.174 | 28.74 / 0.053 | 27.50 / 0.085 | 22.73 / 0.192 | 21.08 / 0.256 | 24.67 / 0.180 | 22.14 / 0.189 | 24.38 / 0.161 |
| DynPoint (K = 6) | **24.69 / 0.097** | **31.34 / 0.045** | **29.30 / 0.061** | **24.59 / 0.086** | **22.77 / 0.099** | **27.63 / 0.049** | **25.37 / 0.039** | **26.53 / 0.068** |

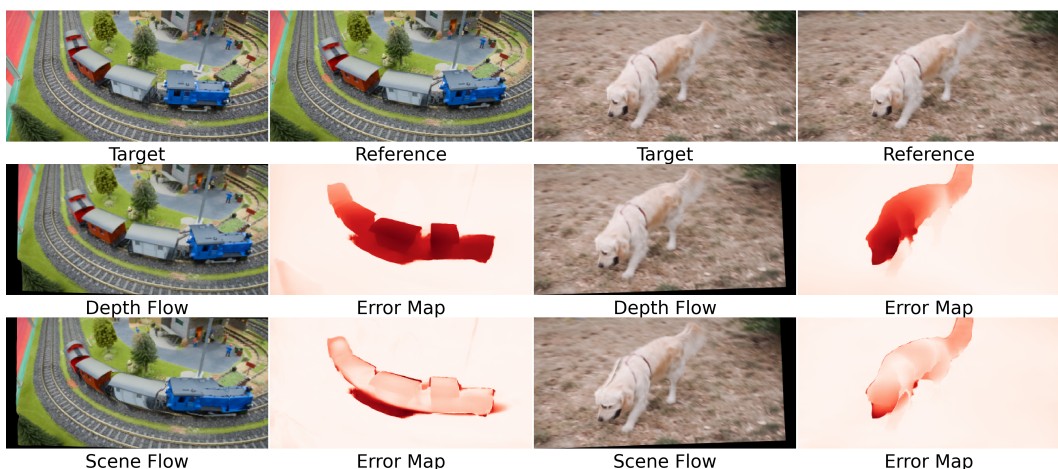

Figure 5: **Demonstration of Depth and Scene Flow Estimation.** This figure presents the output of the target images obtained by warping the reference image using depth estimation (second row) or using both depth and scene flow estimation (third row). It is important to clarify that the figure is not intended for comparing view synthesis results. The synthesized figures generated based on scene flow inherently incorporate object motion as input, resulting in observable motion blur within the synthesized figures. Additionally, an error map represented by the intensity of red is provided to visualize the performance, where deeper shades of red indicate larger errors (in terms of pixel movement compared to corrected optic flow).

scenarios, namely Playground, Skating, Truck, and Umbrella, are presented in Fig. 3. Notably, leveraging monocular depth estimation, DynPoint generally produces views with improved geometric features, as evident in the Skating case shown in the second row of Fig. 3. Even in challenging scenarios like Umbrella, DynPoint successfully generates high-quality views while minimizing blurring effects. **Nerfie** (Tab. 2): In the case of the extended scenario, DynPoint exhibits superior performance in terms of PSNR and achieves comparable results to those obtained in the short video Nvidia, as presented in Tab. 2. This achievement can be attributed to our information aggregation approach, which focuses on effectively aggregating information from the target frame by establishing associations between its points and those in the reference frames. Novel view synthesis results for scenarios TAIL and TOBY SIT are depicted in Figure 4. It is evident that for longer videos, DynPoint continues to produce more realistic frames. Notably, the generated views in this figure do not exist in either the training or test dataset. **HyperNeRF** (Tab. 3): In the case of longer video sequences, DynPoint showcases superior performance in PSNR, as highlighted in Table 3. **Iphone** (Tab. 4): In the context of monocular videos without camera teleportation, as demonstrated in the work [17], DynPoint attains comparable outcomes to previous algorithms. Due to the inherent limitations of having few multi-view perspectives and overlapping information, establishing correspondences between adjacent frames becomes challenging. Consequently, the view synthesis task based on monocular videos proves to be more arduous on this dataset compared to other datasets.

## 4.2 Ablation Studies

In order to evaluate the effectiveness of the strategies proposed in our paper, we conducted three ablation studies. These studies include: (1) the absence of the multiple-step training strategy outlined in Sec. 3.2 and Sec. 3.3; (2) the utilization of only six frames in the vicinity ($K = 3$); and (3) the exclusion of the hierarchical point cloud. The results are presented in Tab. 5. It is evident from the results that the omission of the multiple-step training strategy leads to the largest drop in performance. Without this strategy, the first stage produces noisy outcomes for both monocular depth and scene

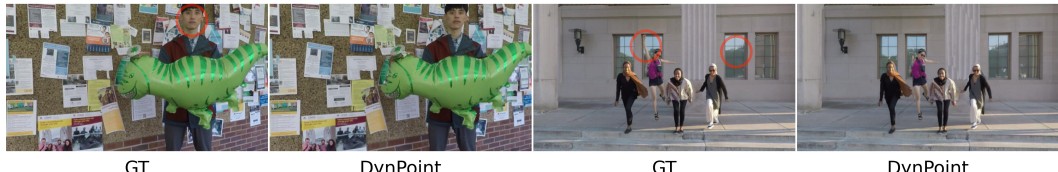

| GT | DynPoint | GT | DynPoint |

Figure 6: **Demonstration of Failure Results on Nvidia Dataset.** In this demonstration, we present the failure results generated by DynPoint.

flow estimation, consequently hindering the generation of the neural point cloud and impacting the performance of the second stage. Moreover, using a limited number of adjacent frames also adversely affects the final performance, which aligns with our expectations as limited inputs correspond to limited information. Although the removal of the hierarchical point cloud does not significantly degrade performance, it still plays a role in generating finer results.

### 4.3 Consistent Depth & Scene Flow Estimation On Davis

In order to validate the efficacy of the monocular depth estimation technique and the scene flow estimation method, we present the following visualization results: (1) Reconstructed Target Image using Monocular Depth Estimation: We demonstrate the reconstruction of the target image by warping the reference image based on the monocular depth estimation. (2) Reconstructed Target Image using Scene Flow and Depth Estimation: We present the visual reconstruction of the target image achieved through warping the reference image using solely depth estimation or a combination of scene flow estimation and depth estimation. The results are displayed in Fig. 5. Upon observing the second row, it is evident that DynPoint demonstrates commendable performance in reconstructing the static background of the target frame through monocular depth estimation. However, the primary errors (deep red part) occur in the dynamic region, where accounting for object movement becomes crucial for accurate warping. Moving to the third row, our scene flow estimation proves to be effective in capturing the movement of dynamic objects, as in the red error map Fig. 5.

### 4.4 Failure Cases

Despite the notable achievements of DynPoint, certain failure cases were observed during the view synthesis process, as demonstrated in Fig. 6. By comparing the first and second images, it becomes apparent that generating realistic facial features in regions with intricate details proves to be challenging. Furthermore, when comparing the third and fourth images, it is evident that DynPoint struggles with handling fine objects and reflections, as these aspects heavily rely on the accurate geometry inference obtained in the first stage.

## 5 Conclusion

In this research paper, we present DynPoint, an algorithm specifically designed to address the view synthesis task for monocular videos. Rather than relying on learning a global representation encompassing color, geometry, and motion information of the entire scene, we propose an approach that aggregates information from neighboring frames. This aggregation process is facilitated by learning correspondences between the target frame and reference frames, aided by depth and scene flow inference. The experimental results demonstrate that our proposed model exhibits improved performance in terms of both accuracy and speed compared to existing approaches.

**Acknowledgements.** Our research is supported by Amazon Web Services in the Oxford-Singapore Human-Machine Collaboration Programme and by the ACE-OPS project (EP/S030832/1). We are grateful to all of the anonymous reviewers for their valuable comments.

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
