# Supplementary Material
# DynPoint: Dynamic Neural Point For View Synthesis

**Kaichen Zhou, Jia-Xing Zhong, Sangyun Shin, Kai Lu,**
**Yiyuan Yang**, **Andrew Markham**, **Niki Trigoni**
Department of Computer Science
University of Oxford
{rui.zhou, jiaxing.zhong, sangyun.shin, kai.lu}@cs.ox.ac.uk
{yiyuan.yang, andrew.markham, niki.trigoni}@cs.ox.ac.uk

## 1   Network Structure

In this section, we present the architectural details of the scene flow MLP and rendering MLP models to facilitate the reproducibility of our experiments. These architectures serve as crucial components in our framework and play a significant role in the overall scene flow estimation and rendering processes. By providing the detailed specifications of these models, we aim to enable researchers and practitioners to reproduce our results effectively and potentially build upon our work for further advancements in the field.

**Scene Flow MLP:** To expedite the inference process, we have devised a formulation for the MLP utilizing a Convolutional Network Structure with the kernel size $(1 \times 1)$. This architectural design choice allows for enhanced computational efficiency and speed during the inference phase. Rather than processing position information $(x, y, z)$ on a point-by-point basis, we adopt a holistic approach by considering the entire image's points as input. These points are concatenated with additional embedded information, including the positional information $e(x, y, z)$, the temporal information $t$, and the embedded temporal information $e(t)$. This comprehensive input configuration allows for a more comprehensive understanding of the scene, incorporating both spatial and temporal cues. By incorporating these additional embeddings, our model gains a richer representation of the data, enabling it to capture intricate relationships and dependencies between points in the image, thereby enhancing its ability to make accurate predictions. The structure of scene flow MLP is shown in the upper part of Fig. 1.

**Rendering MLP:** The rendering MLP, depicted in Fig. 2, takes position $q$ and direction $d$ as inputs to predict density $\sigma$ and color $c$. It employs $M$ points $\hat{P}_t^m$ around radius $R$ of the queried point $q$, with point-wise feature $f^m$, processed by a feature network. The feature network outputs point-wise features $\hat{f}_t^m$, used to compute a weighted summation $\overline{f}_t$ with weights (the product of the inverse distance between $\hat{P}_t^m \& q$ and the confidence score $\gamma_t^m$, which is demonstrated by arrow). This summation is concatenated with direction $d$ and passed through a Color network to compute final color $c$. Simultaneously, $\hat{f}_t^m$ is used in the Density network to predict point-wise density $\sigma_t^m$, which is weightedly summed to obtain the final density $\sigma$. Eqn. 9 utilizes both density $\sigma$ and color $c$ to compute pixel-wise color information $C$. The Feature Net, Density Net, and Color Net consist of three fully connected layers.

## 2   Pose Correction For Colmap

Pose estimation in dynamic scenarios is consistently a challenging endeavor [10, 1]. Our investigation reveals that in scenarios where the precise ground truth pose is not available, and the estimation of transformation pose is dependent on the original COLMAP, there is a possibility of obtaining

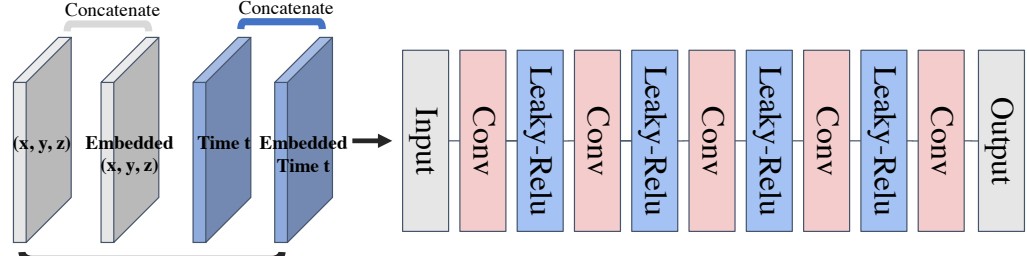

Figure 1: **Structure of Network.** The scene flow network architecture, depicted in the figure, is utilized in the initial stage of our pipeline. It takes 3D position $(x, y, z)$ and time information $t$ as input and predicts both forward and backward scene flow.

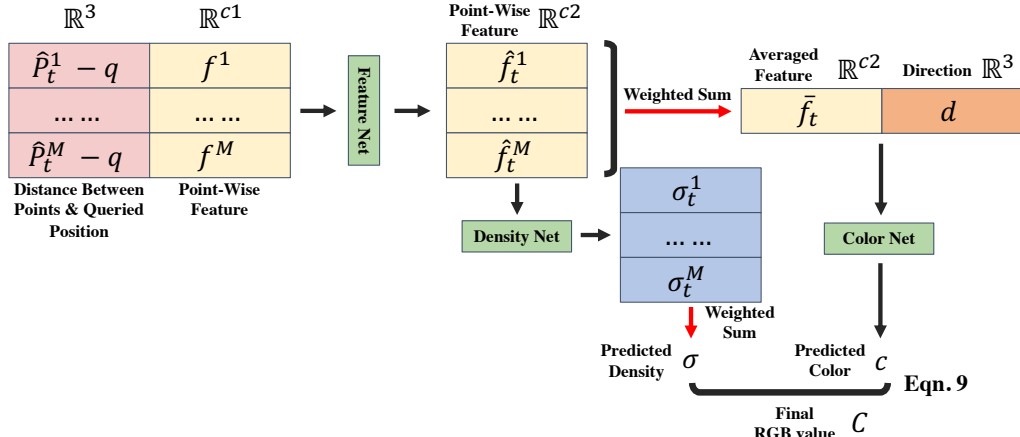

Figure 2: **Structure of Network.** The rendering network architecture, illustrated in the figure, is employed in the second stage of our pipeline. It takes queried points $q$ and direction $d$ as input and predicts the final pixel-wise color $C$.

inaccurate pose estimations. This, in turn, can have a detrimental impact on the effectiveness of our depth and scene flow estimation processes. The optimization of scale parameters, denoted as $\alpha_t \& \beta_t$, relies on the triangulated depth estimation, denoted as $\bar{d}_t$. However, it is important to note that the accuracy of $\bar{d}_t$ is influenced by the accuracy of the camera configuration, represented by the matrix $[\mathbf{R}_{t,c}, \mathbf{t}_{t,c}]$. Despite the incorporation of a motion mask in the COLMAP framework, accurate pose estimation is still a challenge in scenarios characterized by either texture-less environments or small camera motions. These conditions hinder the ability of COLMAP to effectively estimate precise pose information. To enhance the accuracy of pose estimation, we propose a novel algorithm that utilizes a previously estimated mask, represented as $\mathcal{M}_f$.

## 3 Experimental Details

### 3.1 Evaluation Metrics

**PSNR and mPSNR:** Peak signal-to-noise ratio (PSNR) and mean peak signal-to-noise ratio (mPSNR) are widely used evaluation metrics in image and video processing to assess the quality of a processed or reconstructed signal relative to a reference signal. Expressed in decibels (dB), PSNR is calculated by comparing the Mean Squared Error (MSE) between the original and processed signals. The formula for PSNR is as follows:

$$PSNR = 10 * log10((MAX^2)/MSE) \tag{1}$$

Here, MAX represents the maximum possible pixel value of the signal (typically 255 for an 8-bit image), and MSE is the mean squared difference between the original and processed signals.

A higher PSNR value indicates lower levels of distortion or error, indicating better fidelity to the original signal. In essence, a higher PSNR value signifies a higher degree of similarity between the original and processed signals.

**LPIPS:** LPIPS (Learned Perceptual Image Patch Similarity) is a metric used to measure the similarity between images based on perceptual features. It leverages a deep neural network to map image patches to perceptual similarity scores. The process involves preprocessing the images, extracting features using a pre-trained neural network, extracting image patches, computing similarity between corresponding patches, aggregating patch similarity scores, and normalizing the similarity score. LPIPS has demonstrated strong correlation with human perception and finds applications in image quality assessment and image-to-image translation. Its utilization allows researchers and practitioners to evaluate the performance and quality of computer vision algorithms by quantifying perceptual similarity between images.

**mSSIM:** Mean Structural Similarity Index (mSSIM) is an objective image quality metric that measures the similarity between a reference image and a distorted image. It extends the Structural Similarity Index (SSIM) by considering multiple scales of image details. It divides the images into scales, calculates the SSIM index for each scale using luminance, contrast, and structure information, and averages the weighted SSIM index values to obtain the mSSIM value (-1 to 1). Higher mSSIM values indicate better image quality. However, mSSIM may not always align perfectly with human perception, so it is often used in conjunction with other metrics and subjective testing for a more comprehensive assessment.

### 3.2 Datasets

**Nvidia Dataset:** Our method is evaluated on the Dynamic Scene Dataset [9], which comprises 9 video sequences. The sequences are captured using a static camera rig consisting of 12 cameras. Simultaneous images are captured by all 12 cameras at 12 different time steps $\{t_0, t_1, ..., t_{11}\}$. The input consists of monocular videos with twelve frames, denoted as $\{I_0, I_1, ..., I_{11}\}$, where each frame is obtained by sampling an image from the i-th camera at time $t_i$. Notably, each frame of the video is captured using a different camera to simulate camera motion. The frames contain a background that remains static over time, as well as a dynamic object that exhibits temporal variations. To estimate camera poses and the near and far bounds of the scene, we employ COLMAP, following a similar approach to NeRF [5]. We assume a shared intrinsic parameter for all cameras. Following [2, 4], we use Jumping, Skating, Truck, Umbrella, Balloon1, Balloon2 and Playground, 7 scenes for our evluation. Lastly, we resize all sequences to a resolution of $480 \times 270$.

**Nerfie Dataset:** Our method was evaluated on the Nerfie [6]. The dataset employs two methods for data capture: (a) capturing synchronized selfies using the front-facing camera, achieving sub-millisecond synchronization, and (b) recording two videos with the back-facing camera, manually synchronizing them based on audio, and subsampling to 5 frames per second. Image registration was performed using COLMAP [8] with rigid relative camera pose constraints. Sequences captured with method (a) have fewer frames (40-78), but precise synchronization of focus, exposure, and timing. Sequences captured with method (b) have denser temporal sampling (193-356 frames), but less precise synchronization with potential variations in exposure and focus between the cameras. Following the dataset's configuration, we divided the dataset into training and testing sets. The left view was assigned to the training set and the right view to the validation set, and vice versa, in an alternating manner. This allocation ensures that all areas of the scene have been observed by at least one camera, preventing unobserved regions.

**Iphone Dataset:** In this study, we extend the evaluation of our proposed method by applying it to the more challenging Iphone dataset introduced by [3]. This dataset comprises 14 sequences that exhibit non-repetitive motion, encompassing diverse categories such as generic objects, humans, and pets. To conduct our evaluation, we follow the setup in [3] consisting of three cameras for multi-camera capture. Specifically, one camera is handheld and in motion during the training phase, while the other two cameras remain static with a large baseline and are used for evaluation purposes. Additionally, the Iphone dataset provides metric depth information obtained from lidar sensors, which can be utilized for supervising the depth estimation task.

**Davis Dataset:** To test the performance of monocular depth estimation and scene flow estimation module, we also provide the visualization result on the DAVIS (Densely Annotated VIdeo Segmenta-

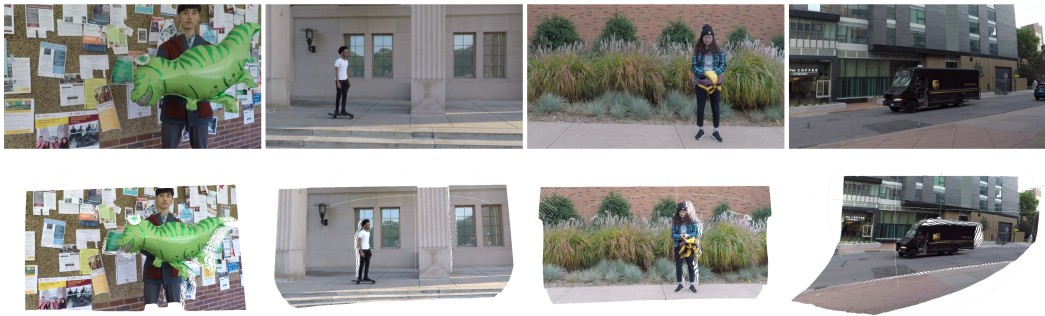

Figure 3: **Point Cloud Based on Depth Estimation.** This figure presents the accuracy of estimated depth, with the upper portion depicting the original images and the lower portion representing the point cloud generated using depth information and camera matrix.

tion) dataset [7], which is a benchmark dataset commonly used in computer vision research for the task of video object segmentation.

## 4   Depth Visualization

In this section, we present the visual point cloud results obtained from the consistent depth estimation module applied to the Nvidia dataset [9]. In our methodology, we incorporate a pose refinement algorithm to enhance the accuracy of pose estimation, which is initially obtained using COLMAP. By utilizing this pose refinement algorithm, we aim to improve the quality of the depth estimation process. The visualization of the point cloud results allows for a comprehensive evaluation of the effectiveness of our proposed approach in achieving accurate and consistent depth estimation on the Nvidia dataset shown in Fig. 3. It is apparent that estimating depth generally allows us to perceive the shape and size of objects in various situations.

## 5   Scene Flow Visualization

This section presents additional visualization results of error maps obtained from the Davis dataset. These results serve to demonstrate the performance of our proposed scene flow module in handling two specific scenarios: (1) the presence of fine details, represented by the train scenario, and (2) the occurrence of motion blur, exemplified by the dog scenario. These visualizations provide a comprehensive understanding of how our method performs in challenging situations where fine details and motion blur are prevalent shown in Fig. 4.

## 6   Difference with NSFF

The NSFF paper has undeniably made a significant contribution to the realm of monocular view synthesis by integrating scene flow into the framework of monocular video-based algorithms. However, there are several differences that set NSFF and DynPoint apart. Firstly, DynPoint introduces a corrective mechanism to adjust the depth scale obtained from monocular videos. This correction greatly improves the consistency of geometry across different frames. Furthermore, NSFF predicts flow for all points (128) along a ray, while DynPoint focuses exclusively on predicting scene flow for surface points. This approach brings two important advantages: firstly, it adds specific constraints to the scene flow of surface points by using predicted optic flow and depth information, which significantly enhances the learning process; secondly, by concentrating on surface points, the scene flow prediction process is made faster. Lastly, unlike NSFF, which encodes information extensively within the MLP, DynPoint gathers information from neighboring frames using 3D correspondence. This unique approach empowers DynPoint to more effectively learn from sequences of long videos.

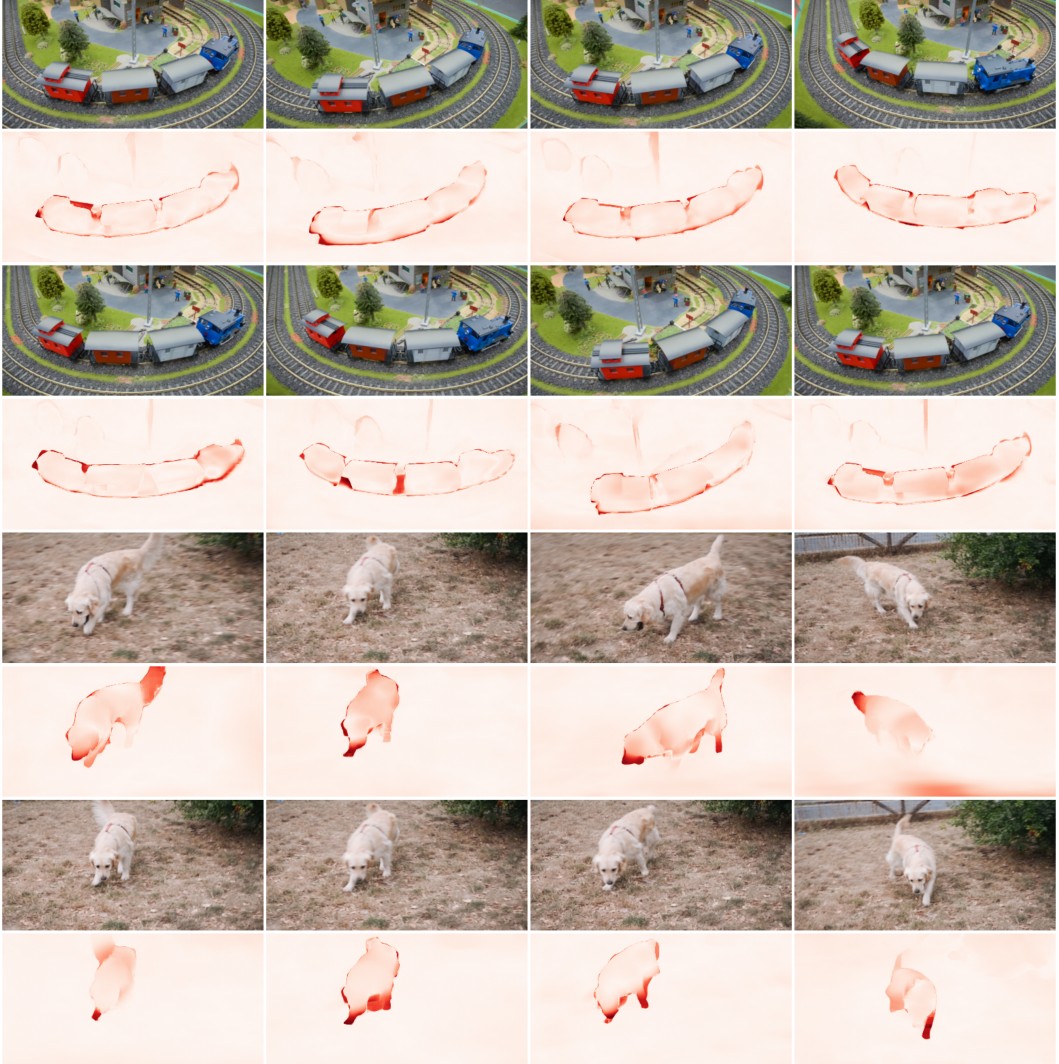

Figure 4: **Demonstration of Scene Flow Estimation.** The odd rows (1, 3, 5, 7) of the presented data exhibit the original images, while the even rows (2, 4, 6, 8) display the error map. Error map represented by the intensity of red is provided to visualize the performance, where deeper shades of red indicate larger errors (in terms of pixel movement based on scene flow compared to optic flow).