# OpenReview forum: "DynPoint: Dynamic Neural Point For View Synthesis"
_NeurIPS.cc/2023/Conference — NeurIPS 2023 poster_

### Official Review · Reviewer_BEKz · 2023-07-02

**Soundness:** 4 excellent
**Presentation:** 3 good
**Contribution:** 3 good
**Rating:** 7
**Confidence:** 4

**Summary:**

DynPoint is an algorithm proposed to enhance the synthesis of novel views in monocular videos using neural radiance fields. It focuses on predicting explicit correspondence between neighboring frames, enabling information aggregation for view synthesis. The demonstrated experimental results show a significant reduction in training time, comparable outcomes to previous approaches, and robustness in handling uncontrolled and lengthy videos without requiring a canonical representation of video content. To achieve it authors propose to improve the estimated scene flow from the optical flow with corrected depth values

**Strengths:**

the proposed technique for scene flow estimation is a solid contribution. It takes into account two inconsistencies in depth and flow that are denoised with the method.  The improvements can be applied to any baseline as a separate technique to improve baseline (e.g. NSFF). Moreover, this allows extracting correspondences for the arbitrary 3D scene.
hierarchical scheme to render the semi-explicit structure is a novel and a trustable way to get the fast method based on existing research
Experiments demonstrate the substantial of the proposed method and provides a strong contribution to the field.



**Weaknesses:**

The proposed method relies heavily on existing models and it is unclear whether the quality depends if you replace them with others
Authors don't investigate the quality of the correspondence, which can be interesting by itself. The easiest way is to extend section 4.4 with a comparison versus method like "Deformable Sprites" Ye et al.
The presentation of the visual results can be improved. Figure 3 should have a close-up by emphasizing the region of interest, otherwise, results are not clear as well as others.
Not compared with “DynIBaR: Neural Dynamic Image-Based Rendering”. Yes, I understand that it is a recent work. At least add numbers to the table for Nvidia dataset.
Without video results, it is difficult to reason even with such metric improvements on average.
No limitation or ethical section to give a reader explicit statements.

**Questions:**

1. How do you measure the time? Does it include phase 1 and the hardware is the same as competitors?
2. “Neural point cloud” technique is similar to [1, 2, 3] representations in some sense (humans & static scene examples). Could you extend related works in the neural point cloud as well? Does hierarchy is an important contribution for the neural point cloud? If so, it can be
3. What is the main drawback of the method for iPhone dataset? Is it possible to extend the comparison on this benchmark with others, since the gap is much smaller than on others?
4. Do you consider including video metrics for the results (e.g. VMAF)?
5. How issues in Figure 5 can be addressed and what is causing them?

[1] PointAvatar: Deformable Point-based Head Avatars from Videos
[2] Self-Improving Multiplane-To-Layer Images for Novel View Synthesis
[3] Pulsar: Efficient Sphere-based Neural Rendering

**Limitations:**

Fast changes in the scene as well as thin objects can be a problem for the method, which is quite common for this class of models. Compared with other methods this one needs more information and additional pretraining steps. the category-specific methods can be better for non-rigid parts. The scalability of the method is limited due to two-stage training for each of the scenes.

---

> ### Author Rebuttal · Authors · 2023-08-09
>
> # Section 4 - Reviewer BEKz
>
> We thank the reviewer for the constructive assessment of our work. In the following, we address the concerns point by point. Please feel free to use the discussion period if you have any additional questions.
>
> ## 4.1 Weakness
>
> **4.1.1 Deformable sprites**
>
> Unlike our initial focus on 3D relationships, "Deformable Sprites" aims to learn a non-rigid 2D pixel-wise transformation using video data, enabling motion separation without supervision. This makes a direct comparison between our learned 3D correspondences and their results challenging. But, both approaches utilize predicted optical flow from RAFT [53] to assist in determining the correspondence between two frames. However, the concept of using video to understand unsupervised motion for generating new views is attractive, which enables specific rules for different motion groups.
>
> **4.1.2 Region of interest**
>
> Certainly, highlighting the key area will improve reader understanding. We'll include this in the final version.
>
> **4.1.3 DynIBaR**
>
> DynIBaR is indeed a relevant study to ours. We'll add their metrics in the final version and we also provide a comparison in the Reviewer Rm9i (Sec. 3.1.1).
>
> **4.1.4 Video results**
>
> We're grateful for your insightful observation. Following the instructions for the rebuttal process, we've added an anonymous link to the video demo we've made for the HyperNeRF dataset. You can find this link in our response to the AC under the "Official Comment" box at the top of the review page. More demos could be found in the PDF document to AC.
>
> **4.1.5 Limitation or ethical section**
>
> Our method's primary limitation lies in its generalizability. Addressing this issue would significantly enhance the feasibility of DynPoint. Further details can be found in the Reviewer J3AP (Sec. 1.2.2).
>
> ## 4.2 Questions
>
> **4.2.1 Measurement of time**
>
> We've tested two stages of DynPoint using the same computer setup.
>
> **4.2.2 Neural point cloud**
>
> Your insights are appreciated. Recent findings in Sec. 4.3 (main text), Ablation Studies, confirm the positive impact of our introduced hierarchical structure on overall performance. The three referenced papers offer interesting viewpoints on addressing view synthesis challenges.
>
> In reference to [1], it highlights neural points for video-based Head Avatars reconstruction. Our approach reconstructs point clouds from refined depth and predicted scene flow, differing from [1]'s emphasis on direct learning of canonical point cloud representation. Our research seeks a universal solution for monocular video-based view synthesis tasks, diverging from [1]'s specific focus. Implementing our hierarchical structure might enhance [1]'s approach.
>
> Regarding [2] and [3], their strategies differ from our neural point methodology. [2] introduces front-parallel planes for static scenarios, and [3] proposes spheres for a similar purpose.
>
> **4.2.3 iPhone dataset**
>
> The smaller gaps in the iPhone dataset are due to its limited information from various angles, as indicated by "effective multi-view factors" proposed by DynCheck [17]. This dataset naturally has fewer diverse camera views, making the reconstruction of appearance more challenging compared to the Nvidia and Nerfie.
>
> **4.2.4 Video metrics**
>
> Your question prompted us to consider a new perspective. We've realized that VMAF has broader applications beyond creating new views from videos. It can be useful for evaluating view synthesis in static scenarios as well. It's intriguing that this aspect hasn't been widely explored in existing research. Therefore, we acknowledge the significance of reevaluating and thoroughly examining the suitability of VMAF's features for view synthesis tasks.
>
> **4.2.5 Figure 5**
>
> Figure 5 highlights three failure instances: facial expression, reflection-induced, and fine object view synthesis. Facial expression reconstruction faces challenges due to humans being sensitive to these artifacts. Models like 3DMM improve this aspect.
>
> Reflections can change surface appearances, making it hard to distinguish actual geometry from reflections. Illumination and viewing angles further complicate achieving consistent appearances. Advancements like "NeRFReN" Guo, Yuan-Chen, et al. integrate reflections for better results.
>
> View synthesis for fine objects or high-resolution images is an active research area. Innovative architectures, such as "3D Gaussian Splatting for Real-time Radiance Field Rendering," are designed to solve it.
>
> ## 4.3 Limitations
>
> **4.3.1 Fast change \& Thin object**
>
> Indeed, dealing with fast changes and thin objects in scenes poses challenges. Fast changes mean less matching between frames, and thin objects involve complex geometry. Both of these aspects make monocular-video-based view synthesis more difficult.
>
> **4.3.2 Pretraining steps**
>
> Yes, the initial pretraining is vital for our approach. However, we use fewer learnable weights, leading to shorter total training times than other methods.
>
> **4.3.3 Category-specific method**
>
> It's true that incorporating category-specific details can improve the performance of view synthesis models in specific scenarios, e.g., SMPL for human reconstruction and 3DMM for facial expression reconstruction. Our paper focuses on introducing a generalizable algorithm that doesn't rely on specific prior assumptions about objects.
>
> **4.3.4 Scalability**
>
> Scalability, ensuring optimal performance with bigger and more complex tasks or datasets, is a consideration for our model. This affects our model's initial stage in DynPoint, involving explicit correspondence learning. But it's worth noting that other one-stage models face similar hurdles. Our experiments with the iPhone Dataset revealed performance declines across models. One-stage approaches like NSFF aim for a universal video representation, which requires implicit frame correspondence. However, the lack of explicit correspondence adds complexity to implicit correspondence learning.

---

> > ### Comment · Reviewer_BEKz · 2023-08-21
> > **Reply**
> >
> > I want to thank the authors for their effort in addressing my concerns. After reading their response and considering the feedback from other reviewers, I believe that the new experiments provide a comprehensive evaluation of their proposed approach for me, and I maintain a positive view on its acceptance.

---

> > > ### Author Response · Authors · 2023-08-21
> > > **Response to Comments**
> > >
> > > Dear reviewer BEKz,
> > >
> > > We extend our sincere appreciation for your invaluable and constructive suggestions. Your feedback has played a crucial role in enhancing the final version of our paper. Your insights and recommendations have greatly contributed to the refinement of our work. Thank you for your time and effort in providing such insightful comments, which have undoubtedly strengthened the quality of our research.
> > >
> > > Best regards,

---

> ### Comment · Area_Chair_m7ys · 2023-08-20
>
> Dear Reviewer,
>
> Thanks for your valuable comments. Would you please have a look at the authors' rebuttal and other reviewers' comments and share you comments here? Thanks!

---

### Official Review · Reviewer_Rm9i · 2023-07-03

**Soundness:** 4 excellent
**Presentation:** 4 excellent
**Contribution:** 3 good
**Rating:** 7
**Confidence:** 4

**Summary:**

The paper proposes a novel method for performing novel view synthesis from monocular captured videos. This is done by learning scene flow and depth parameters which enable the accurate aggregation of appearance information from nearby frames. The paper proposes a novel method of acquiring both consistent depth estimations for a video, and consistent scene flow represented by an MLP. These two sources of information are used to synthesize novel views at a specific time from an arbitrarily specified viewpoint by sampling from corresponding information in a window of nearby frames. The proposed method is demonstrated to outperform existing methods based on representation learning in synthesizing novel views from monocular videos, and this is shown for a wide variety of datasets.

AFTER REBUTTAL:

I have read the author rebuttal, and believe that it has addressed my questions. I appreciate the detailed discussion on the comparison to DynIBaR. I also believe the additional videos provided have definitely helped improve my opinion of the method, although I wish the method was tested with more qualitative video examples (as mentioned by another reviewer). However, I believe the testing now is sufficient.

**Strengths:**

In my opinion, the main strengths of the paper are that:
1. The presentation of the paper is concise, clear, and thorough.
    - The introduction and related work I find to be comprehensive, and accurately describe what the contributions are and what the current issue is with the practice.
    - The methods section is described in detail, and is relatively clear to understand and follow.
    - The figures are all informative and demonstrate the capabilities of the method. One thing I wish could be added is a video result demonstrating consistent novel view synthesis.
2. The proposed contributions are original and impactful.
    - The module for estimating consistent depth for a monocular video seems novel, and while there is existing work tackling this (see questions), the method proposed seems to generate good enough results for novel view synthesis with DynPoint.
    - The method for estimating and representing the scene flow between adjacent frames and using this to aggregate information from reference to target frames seems novel (although based on image-based rendering ideas), and learning this from the video seems like an important contribution.
3. The evaluations are sound.
    - The evaluations are done for many (4) different datasets, and compare to many different dynamic scene representation methods (Nerfies, D-NeRF, NSFF, HyperNeRF among others). Because of how thorough these evaluations are, and DynPoint outperforms all of the baseline methods, it is very convincing that the proposed method is robust and is able to generate high-quality results.
    - The ablations are relatively thorough, and Table 4 shows how the individual parts of the representation and aggregation pipeline contribute to the final quality. I found it to be informative about the method.

**Weaknesses:**

I do not view there to be many major weaknesses of the paper. One thing which would be interesting is a comparison to the method DynIBaR [31]. The paper mentions that this method is limited to short videos due to the capacity of MLPs or other representation mechanisms (L34), but from my understanding this method uses a similar philosophy as this work where new frames are synthesized as inspired by image-based rendering and scene flow estimation. This seems to me to be the current state-of-the-art. Additionally, it’s not clear that the videos this method is evaluated on are long enough for this to be a concern, considering NeRF-based methods can be evaluated and these are constrained by the capacity of the MLP being used to represent the scene. Additionally, I think there could be a bit more discussion on the limitations of the proposed method (see limitations).

**Questions:**

- How does the generated consistent video depth estimation compare to other work in this field, for example: https://roxanneluo.github.io/Consistent-Video-Depth-Estimation/. Could this method be dropped in and used for this estimation step?

**Limitations:**

The paper adequately addresses the limitations of the method. One additional limitation that discussion of would significantly improve the quality of the paper, is the amount of differing from the monocular video path which can be taken and still synthesize reasonable novel views. Obviously, since this method is using appearance information in nearby frames, it is likely to not be able to hallucinate information beyond these. However, NeRF-based methods are able to put something there and potentially reconstruct better results due to the smoothness of the representation learned by the MLP. Some comparison of the existing methods and the proposed method on this axis (range of possible novel views synthesized) would be helpful, and if this is a limitation of the proposed method then it should be discussed.

---

> ### Author Rebuttal · Authors · 2023-08-09
>
> # Section 3 - Reviewer Rm9i
>
> We thank the reviewer for the constructive assessment of our work. In the following, we address the concerns point by point. Please feel free to use the discussion period if you have any additional questions.
>
> ## 3.1 Weaknesses
>
> **3.1.1 Comparison with DynIBaR**
>
> This is an insightful observation. Indeed, our perspectives align closely, and we are concurrent work. Nevertheless, discernible distinctions between DynPoint and DynIBaR exist, serving to underscore the robustness of the proposed DynPoint methodology.
>
> Both DynIBaR and DynPoint harness information aggregation mechanisms to realize the synthesis of novel views. However, DynIBaR predominantly centers around the aggregation of information through two-dimensional (2D) pixel units. This approach draws inspiration from image-based rendering principles, entailing the synthesis of novel perspectives from a collection of reference images via a weighted fusion of reference pixels. In contrast, DynPoint's focal point lies in the information aggregation achieved by constructing three-dimensional (3D) neural point clouds. The final novel view synthesis is realized by using 3D neural points surrounding the queries' position.
>
> Moreover, DynPoint introduces an efficacious strategy for the seamless integration of monocular depth estimation within the ambit of monocular video view synthesis. In contrast to DynIBaR, which endeavors to model the trajectory of all samples (128) traversing each ray, DynPoint exclusively focuses on the trajectory of surface points, thereby yielding a substantial acceleration in both the training and inference stages.
>
> **3.1.2 Long video**
>
> We value your insight. To address this, we examined our model using the iPhone dataset, encompassing numerous images (several hundred frames) across different scenarios in comparison to Nerfie and Nvidia datasets. Our model demonstrated stable performance within this context. However, we acknowledge a minor improvement in the iPhone dataset compared to the significant enhancements seen in Nvidia and Nerfie datasets. This discrepancy can be attributed to the limited information available from various angles in the iPhone dataset, as indicated by the "effective multi-view factors" proposed by DynCheck [17].
>
> To further explore DynPoint's capabilities with longer videos, we followed Review J3AP's suggestion and conducted relevant experiments on the HyperNeRF dataset in the Reviewer J3AP (Sec. 1.1.3), which also contains hundreds of frames per scenario. The results and our algorithm's proficiency in handling longer videos are presented in the attached PDF document. For clarity, we’ve also included a subset of results in the reviewer J3AP (Sec. 1.1.3).
>
> ## 3.2 Questions
>
> **3.2.1 Consistent-Video-Depth-Estimation**
>
> Thank you for bringing this up. It's a notable point, as the paper you mentioned indeed provides valuable insights. Both our methods draw inspiration from traditional structure-from-motion reconstruction principles to establish the connection between monocular depth and optical flow. It's possible that their algorithm could be integrated into our depth estimation process.
>
> However, there are distinct differences between our DynPoint approach and theirs. While their focus is primarily on achieving accurate depth estimation, which is also a key aspect of our work. Our method focuses on estimating scene flow and depth information together.
>
> Another difference lies in the requirements of our methods. While their approach involves fine-tuning a complex convolutional neural network-based depth estimation architecture tailored for each scenario, our method calls for fine-tuning a simpler scene flow MLP structure and learnable scale factors of the depth map. This divergence significantly reduces the computational demands during the optimization process.
>
> ## 3.3 Limitations
>
> **3.3.1 Limitation concerning global information**
>
> Thank you for bringing this up. Initially, we were concerned about the extent to which global information might impact our method. Unlike approaches such as NSFF or Nerfie, which capture comprehensive scene details for predictions, our method showed promising performance on the iPhone dataset, which contains multiple frames for each scenario. Additionally, the newly added experimental results on the HyperNeRF dataset during the rebuttal period consistently demonstrate the improvement achieved by our approach. The reason for this success might be that even though we only use information from nearby frames to predict stuff, we actually train our scene flow (MLP structure) and render (MLP structure) parts using the whole video, which encodes some global information across frames into our model and empowers our model to make a reasonable inference. More demos on Nerfies and iPhone datasets could be found on the PDF document of the rebuttal to the Associate Chair (AC) at the top of the review page. The video link of HyperNeRF could be found in our official comment.
>
> We also experimented with our model using different numbers of nearby frames, and we explain these results in our ablation studies. What we found is that when we use fewer frames, our model doesn't work as well. But when we increase the number of frames up to a certain point, the improvement becomes less noticeable. One reason for this could be that keeping track of things over a long time doesn't always work perfectly. Another reason could be that the camera and objects move in ways that are hard to predict, so having a longer video might not necessarily give us much more useful information.

---

> > ### Comment · Reviewer_Rm9i · 2023-08-17
> >
> > I appreciate the detailed response and clarifications on my questions. I do not have any additional questions.

---

> > > ### Author Response · Authors · 2023-08-21
> > > **Response to Comments**
> > >
> > > Dear reviewer Rm9i,
> > >
> > > We would like to express our gratitude for the valuable and constructive suggestions you provided. Your input has been instrumental in enhancing the quality of our paper's final version. Your thoughtful feedback has significantly contributed to the refinement of our research. We genuinely appreciate the time and effort you dedicated to offering such insightful comments, which have undoubtedly enriched our work.
> > >
> > > Warm regards,

---

### Official Review · Reviewer_nc3b · 2023-07-04

**Soundness:** 2 fair
**Presentation:** 2 fair
**Contribution:** 1 poor
**Rating:** 3
**Confidence:** 4

**Summary:**

This paper proposes an algorithm for novel-view synthesis in dynamic scenes. According to the abstract and introduction section, this paper takes a monocular video as an input (L.98) and cope with uncontrolled or length scenarios (L.3). Using geometric priors, such as monocular depth and optical flow from off-the-shelf methods, this paper exploits the pixel-wise correspondence to encode sceneflow within the network.

**Strengths:**

Overall, I am a bit not that positive for this paper. Let me put more comments on the weakness section.

**Weaknesses:**

_1. Wrong experimental setup._

This paper claims that they focus on rendering scenes from a monocular video to handle lengthy video. However, this is not true. NVIDIA dataset [63] captures the scenes using a camera rig (2x8 array camera? not sure), which is not a monocular video. I think that the correct meaning of the monocular video is addressed by the Dycheck dataset [17] which is called iPhone dataset [17] in this manuscript. Moreover, NVIDIA dataset [63] only has a few training frame which is not well aligned with the addressed points by the authors in the abstract and introduction. At least, HyperNeRF dataset [42] provides more than 200~ frames per scene, where I could say this is a bit lengthy video. Such inconsistency in addressed problems and the proposed solution is quite not good. I am not that satisfied with the quality of the writing.

Moreover, where can I find the qualitative results on Dycheck dataset [17]? While table 3 provides the quantitative results, I could not find any qualitative results.


_2. No video demo_

More seriously, though this paper tackles the neural rendering in dynamic scenes, I could not find any video demo even in the supplementary material. __Lack of video demo is a big problem__.


_3. What is the difference in technical contribution in comparison with NSFF [30]?_

NSFF [30] also exploit the sceneflow understanding within the network, similar to this paper. What is the main difference? At least authors mainly discuss this issue and provide some quant/qual difference within their experimental section. But I could not find any difference or novelty in this part.


_4. Lack of related papers to address the speed in dynamic NeRFs._

This paper is not the pioneering work that provides fast rendering/training speed in dynamic NeRF. I hope that the authors to look at the paper of TiNeuVox, Siggraph Asia 2022 (https://github.com/hustvl/TiNeuVox). If this paper wants to claim the novelty in speed, the authors should have fairly compared with TiNeuVox.



**Questions:**

I hope that the authors properly address my concerns which are listed in the weakness section.

Overall, this paper does not provide the whole information that is necessary for the reviewers to properly judge its validity.


**Limitations:**

It's okay to me.

---

> ### Author Rebuttal · Authors · 2023-08-09
>
> # Section 2 - Reviewer nc3b
>
>
> We thank the reviewer for the constructive assessment of our work. In the following, we address the concerns point by point. Please feel free to use the discussion period if you have any additional questions.
>
> ## 2.1 Weakness
>
> **2.1.1 Wrong experimental setup**
>
> Weakness Regarding Nvidia Dataset: Your observations are greatly appreciated. We agree with your assessment regarding the limitations of the Nvidia dataset, particularly within the context of monocular video-based view synthesis, a facet also addressed by Dycheck [17]. Despite this, our utilization of the Nvidia dataset is driven by its widespread adoption across related literature in this domain. This deliberate choice enhances the comparability of our algorithm with other approaches.
>
> Inquiries Regarding HyperNeRF Dataset: We extend our gratitude for bringing this matter to our attention. The performance of our algorithm on the HyperNeRF dataset has been incorporated into our comprehensive response, available in the appended PDF document of "global" response. These newly added experimental results on the HyperNeRF dataset consistently demonstrate the improvement achieved by our approach. For clarity, we’ve included a subset of results in the reviewer J3AP (Sec. 1.1.3). The video link of this dataset could be found on "Official Comment" to Associate Chair (AC).
>
> Inquiries Regarding Dycheck (iPhone) Dataset: We appreciate your keen observations. Our response includes the results produced by our algorithm when we tested it on the iPhone dataset. It's important to note that the iPhone dataset presents challenges, and our algorithm struggles to generate clear views in most situations. The demo for Dycheck (iPhone) and Nerfies Dataset could be found on the PDF document of rebuttal to Associate Chair (AC) at the top of the review page.
>
> **2.1.2 Video demo**
>
> We're grateful for your insightful observation, and we recognize how a video demo could really make our paper clearer. Following the instructions for the rebuttal process, we've added an anonymous link to the video demo we've made for the HyperNeRF dataset. You can find this link in our response to the Associate Chair (AC) under the "Official Comment" box at the top of the review page.
>
> **2.1.3 Difference with NSFF**
>
> We want to express our gratitude for your perceptive observation. The NSFF paper has undeniably made a significant contribution to the realm of monocular view synthesis by integrating scene flow into the framework of monocular video-based algorithms. However, there are distinct differences that set NSFF and DynPoint apart.
>
> Firstly, DynPoint introduces a corrective mechanism to adjust the depth scale obtained from monocular videos. This correction greatly improves the consistency of geometry across different frames.
>
> Furthermore, NSFF predicts flow for all points (128) along a ray, while DynPoint focuses exclusively on predicting scene flow for surface points. This approach brings two important advantages: firstly, it adds specific constraints to the scene flow of surface points by using predicted optic flow and depth information, which significantly enhances the learning process; secondly, by concentrating on surface points, the scene flow prediction process is made faster.
>
> Lastly, unlike NSFF, which encodes information extensively within the MLP, DynPoint gathers information from neighboring frames using 3D correspondence. This unique approach empowers DynPoint to more effectively learn from sequences of long videos.
>
> **2.1.4 Comparison with TiNeuVox**
>
> We want to acknowledge your input, which is truly valuable. In our submitted work, we've referenced TiNeuVox and carried out a comparison of its performance with our method using the Nvidia dataset. TiNeuVox is a notable effort that aims to create a more effective way to represent geometry, appearance, and motion information in dynamic situations. While it has its strengths, there are noticeable differences between DynPoint and TiNeuVox.
>
> Firstly, TiNeuVox doesn't explicitly learn about scene flow. Instead, it adopts a method that learns about motion in an indirect manner through a specially designed voxel-style representation. This implicit way of representing motion can be challenging to control effectively during training, which can contribute to its decreased performance in the Nvidia dataset.
>
> Secondly, TiNeuVox tries to capture the entire dynamic scenario within an improved representation. However, this comprehensive approach can face difficulties when dealing with long videos, as it might require more computational resources and memory due to its complexity.
>
> Lastly, we've included a speed comparison in Table 1 of the main text. From this table, we could notice that our model performs faster and achieves higher accuracy when compared to TiNeuVox on the Nvidia dataset.

---

> ### Comment · Area_Chair_m7ys · 2023-08-17
>
> Dear Reviewer,
>
> Thanks for your valuable comments. Would you please have a look at the authors' rebuttal and other reviewers' comments and see whether your concerns have been addressed or not?

---

> ### Comment · Reviewer_nc3b · 2023-08-17
> **Comments to Authors**
>
> Thank you for the detailed response. While the rebuttal was clear and to the point, I remain __skeptical__ about the presented contributions and experimental results of this submission.
>
> ### 1. Single Video Demo?
>
> I checked the quality of the video demo, but the authors __only provide the easiest case__ with only a single scene (Chicken in the HyperNeRF dataset). Why haven't the authors shared comprehensive qualitative results? Considering this paper emphasizes the quality of the rendered video, it's essential to closely examine its temporal consistency. Based on the content shared by the authors, it's challenging for me to assess if the paper genuinely substantiates its statements on dynamic neural rendering quality and enhanced temporal coherence.
>
> - __In my own trials with TiNeuVox and rendering a scene from a static viewpoint, the outcome appears more consistent than what's demonstrated in this paper.__ Note that the authors do not compare the results by TiNeuVox in the video demo.
>
> - Additionally, it's baffling why the comparisons in the video demo are __restricted to just one paper, NSFF__. The current approach makes me __suspicious__ of the quality of this submission. This paper is not ready for the publication.
>
> ### 2. Lack of experiments to support authors' claims
>
> The experiments, in their current form, don't offer significant insights. Upon revisiting the abstract, the authors mention:
>
> - we propose DynPoint, an algorithm designed to facilitate the __rapid__ synthesis of novel views for unconstrained monocular videos
> - our method exhibits strong robustness in effectively handling __long-duration videos__ without learning a canonical representation.
>
> In the authors' position, I would prioritize contrasting our findings with TiNeuVox, which boasts quicker training and rendering in dynamic environments. Even though the authors contend that TiNeuVox doesn't utilize sceneflow (potentially explaining the limited comparative data throughout the paper, supplementary material, and rebuttal), I firmly believe there's a need to set TiNeuVox against all datasets, showcasing full qualitative video results. Without this, __it feels like the authors may have overclaimed the strength of the submission about the rapid rendering.__
>
> Regarding longer video scenarios, reviewer J3AP has similarly noted: _"The paper claim that the method could handle long-duration video, which I agree. However, no experiment is conducted to prove this ability."_ I also checked the additional results that the authors uploaded, but __none of them still proves its strength regarding the lengthy-scenario, in my opinion__.
>
> Here's the key.
>
> - If the authors argue that quantitative metrics are higher than those of previous methods, I would say that it looks trivial to me.
>
> - However, if the authors could provide a video demo similar to DynIBaR presented on their project page, I will absolutely vote for acceptance.
>
> As a reviewer, showing the proper qualitative results is also one important factor to judge the acceptance of the submission.
>
> ### 3. Lack of analysis on the usage of consistent depth
> It quite makes sense to me that the authors extend the monocular geometry cues for learning the sceneflow. Typically, as the other reviewers also commented, using consistent depth estimation is a meaningful approach to the dynamic NeRF setup. However, it is unclear whether using consistent depth is quite effective. In terms of quantitative results, I found the one. However, in terms of qualitative results, I have no idea. This paper targets _not the static scene but the dynamic scene_. So many ablation studies simply provide the number. I am not that satisfied with such a submission.
>
> _Moreover, if the authors want to claim a clear difference compared to NSFF or the novel contribution, the authors should have provided a specific ablation study to alleviate such concerns._
>
> For example, MonoSDF (Song et al., Neurips 2022) also exploits the monocular geometric cues, such as depth maps and surface normal maps. As you can see in the manuscript, it provides tons of qualitative results with detailed ablation studies that strongly supports the authors claim on the benefit of using the monocular cues.
>
> However, in this paper, even though the quantitative results achieve state-of-the-art performance, I believe that showing qualitative results is much persuasive, typically for the dynamic neural rendering task. I hope that the authors could provide bunch of the qualitative results with clear comparison with various baselines. Otherwise, I would say that this paper is not ready for the publication.

---

> ### Comment · Reviewer_nc3b · 2023-08-17
> **To all reviewers and ACs**
>
> I found that the other three reviewers voted for acceptance at the initial rating and I am the only one who is against this paper. If my argument looks too aggressive, let me stop complaining about the contributions of this paper. However, as a reviewer, I cannot agree with this paper at all. I hope that most of the concurrent dynamic NeRF papers present the various and numerous qualitative __video demos__ that strongly support the strength of the contributions.
>
> Let me provide the URLs for the recently-published papers about dynamic NeRF. If the authors provide the proper qualitative results. Let me re-evaluate my scores. If not, I am really not that positive at this moment.
> - [NSFF video demo](https://www.cs.cornell.edu/~zl548/NSFF/)
> - [Space-time Neural Irradiance Fields for Free-Viewpoint Video](https://video-nerf.github.io/)
> - [D-NeRF](https://www.albertpumarola.com/research/D-NeRF/index.html)
> - [Dynamic View Synthesis from Dynamic Monocular Video](https://free-view-video.github.io/)

---

> > ### Author Response · Authors · 2023-08-20
> > **Demo video2**
> >
> > Dear Reviewer,
> >
> > We appreciate your feedback. Kindly find our second demonstration video provided above. Should you have any additional inquiries, please don't hesitate to discuss here.
> >
> > Best regards,
> >
> > Authors of Paper 724

---

> ### Author Response · Authors · 2023-08-17
> **More Demo Videos**
>
> Hello ACs & Reviewers,
>
> We really appreciate your feedback. We know that including more videos can make it easier for you to understand our paper. So, we're working on adding more videos that compare our method with other advanced techniques. As soon as we have it ready, we'll share the link to our new demo video in the official comment box for the ACs.
>
> Thanks again,
>
> Authors of Paper 724

---

> > ### Author Response · Authors · 2023-08-21
> > **Response to Comments**
> >
> > Dear Reviewer,
> >
> > Your feedback on our paper is greatly appreciated. We have taken careful note of your comments, and in upcoming iterations, we are committed to enhancing the quality of our demo. Additionally, we are planning to unveil our project webpage concurrent with the conference publication of our paper. This platform will feature a compilation of all the demos we showcased during the rebuttal process, including both **demo1 and demo2 as outlined in our comments to the Associate Chairs**.
> >
> > Warm regards,

---

### Official Review · Reviewer_J3AP · 2023-07-06

**Soundness:** 3 good
**Presentation:** 3 good
**Contribution:** 3 good
**Rating:** 6
**Confidence:** 5

**Summary:**

The paper proposes a method for dynamic scene synthesis from monocular video. The authors aim to speed up training and deal with long-duration videos. To do this, they propose to estimate scene flows of surface points supervised by the signals which are generated by the proposed consistent depth algorithm. Then the information is aggregated to form a neural point cloud and novel view images are rendered by sampling amount this point cloud.

**Strengths:**

1.	The paper proposes a novel solution to dynamic scene. Instead of estimating density and color of the scene for each time, the paper explicitly warps the pixel features of reference images to target time according to the explicitly reconstructed 3D geometry. This would reduce the time for training, as there less learnable components.
2.	The experiment shows SOTA results on Nvidia dataset.


**Weaknesses:**

1.	Please consider to rewrite the Equation 3.
2.	The results on Nerfie dataset and Iphone dataset are less consistent compared to Nvidia dataset. Why not test TiNeuVox on Nerfie and Iphone dataset? Or test the proposed method on HyperNeRF dataset?
3.	Please highlight all best results in Table 2 & Table 3.
4.	The paper claim that the method could handle long-duration video, which I agree. However, no experiment is conducted to prove this ability.


**Questions:**

1.	What is the render speed of the method, considering the relative complex neural point cloud construction process.
2.	An open question: is it possible to extend this method to a generalizable method, which could generalize to new scenes without training? For example, replace scene flow MLP with an un-learnable method and use a generalizable renderer.
3.	Equation 7 is confused. The warped 2K point clouds are summed or combined?


**Limitations:**

The authors addresed the limitation I concerned, which is the potential failure caursed by the flaws in explicit depths and flows.

---

> ### Author Rebuttal · Authors · 2023-08-09
>
> # Section 1 - Reviewer J3AP
>
>
> We thank the reviewer for the constructive assessment of our work. In the following, we address the concerns point by point. Please feel free to use the discussion period if you have any additional questions.
>
> ## 1.1 Weakness
>
> **1.1.1 Equation 3**
>
> Thank you for pointing it out. The formulation will be adjusted to incorporate the predefined mathematical symbols as follows:
> $s_{t-t'}(p_{t})= \hat{d_{t'}}(p_t+f_{t-t'}(p_t)) R_{t'} K_{t'}^{-1}  (p_t+f_{t-t'}(p_t)) + t_{t'} - P_t.$
>
> **1.1.2 Performance on iPhone dataset**
>
> Thank you for your careful assessment. The smaller gaps in the iPhone dataset are due to its limited information from various angles, as indicated by "effective multi-view factors" proposed by DynCheck [17]. This dataset naturally has fewer diverse camera views, making the reconstruction of appearance more challenging compared to the Nvidia and Nerfie.
>
> **1.1.3 Tested on HyperNeRF**
>
> We greatly appreciate your reminder regarding HyperNeRF's significance as a pivotal benchmark for evaluating monocular video view synthesis algorithms. We have indeed included an exposition of our algorithm's performance on this dataset within the "global" response PDF document. For clarity, we've included a subset of results here.
>
> ### Novel View Synthesis Results of HyperNeRF Dataset
>
> |  PSNR ↑ / LPIPS ↓                   | Broom | 3D Printer | Chicken | Expressions | Peel Banana | Average |
> |----------------------------|--------------|--------------|--------------|---------------|---------------|----------------|
> | Hyper-NeRF        | 20.60 / 0.613      | 21.40 / 0.212            | 27.60 / 0.108       | 22.00 / 0.196            | 24.30 / 0.170            | 23.20 / 0.260 |
> | NSFF        | 26.10 / 0.284      | 27.70 / 0.125            | 26.90 / 0.106       | 26.70 / 0.157            | 24.60 / 0.198            | 26.40 / 0.174 |
> | DynPoint          | **27.40 / 0.248**       | **27.60 / 0.163**            | **28.10 / 0.089**       | **27.90 / 0.147**            | **26.50 / 0.129**            | **27.50 / 0.155** |
>
>
> **1.1.4 Table 2 \& Table 3**
>
> Certainly, highlighting all the best results will improve reader understanding. We will ensure that all the best results are emphasized in our final version.
>
> **1.1.5 Long video**
>
> We acknowledge your astute observation. Owing to the constraints imposed by length considerations, the answer could be found for the second weakness posed by reviewer Rm9i (Sec. 3.1.2).
>
>
> ## 1.2 Questions
>
> **1.2.1 Rendering speed**
>
> Indeed, while our emphasis does not lie in rendering speed optimization, our method outperforms standard NeRF-based approaches in this regard. Demonstrating around 2.78-fold acceleration relative to D-NeRF, our technique capitalizes on the point cloud representation to efficiently bypass unoccupied space. Furthermore, the construction of the neural point cloud is remarkably swift, benefiting from the scene flow computation focusing on surface points. This stands in contrast to the computational demands of D-NeRF and NSFF, which necessitate the prediction of motion for all sampled points (192 samples for D-NeRF and 128 samples for NSFF) along the ray originating from each pixel.
>
> **1.2.2 Generalizability**
>
> Thank you sincerely for raising this valuable point. Designing a generalizable model for a monocular-video-based view synthesis task is always challenging. Just as you said, to make DynPoint generalizable, we also need a generalizable scene flow estimation model. We are currently dedicating our efforts to devising a generalizable approach for estimating scene flow without relying on per-scene fine-tuning in our upcoming work.
>
> Nevertheless, we acknowledge that monocular scene flow estimation presents a formidable challenge due to occlusion caused by both the moving camera and dynamic objects, which can hinder accurate estimation. In this context, we have found significant insights from the prevalent usage of diffusion models in 3D scenarios [69,70], which realize generalizable monocular 3D reconstruction. We hope that the development of such a generalizable scene flow estimation model will greatly enhance the applicability of our method,  which could serve as a direct replacement for our current first stage.
>
> [69] Zero-1-to-3: Zero-shot One Image to 3D Object.
>
> [70] One-2-3-45: Any Single Image to 3D Mesh in 45 Seconds without Per-Shape Optimization.
>
> **1.2.3 Equation 7**
>
> Thank you for bringing up this crucial observation. The central concept of our paper is aggregating information from adjacent frames to improve the inference of appearance and geometry information for the current frame. While both summation and combination methods could potentially be utilized for information aggregation, our experiments reveal that summation is not feasible in practice. This is primarily due to the challenge of identifying two perfectly overlapping points in the 3D space with predicted scene flow, which makes the summation unreliable.
>
> To address this concern, we have opted for the combination method, which effectively densifies the neural point cloud of the current frame by integrating information from nearby frames. This approach offers a viable solution to achieve information fusion without encountering the issues associated with direct summation.

---

> > ### Comment · Reviewer_J3AP · 2023-08-14
> >
> > Thanks for addressing my concerns.

---

> > > ### Author Response · Authors · 2023-08-14
> > >
> > > Dear reviewers, we appreciate your valuable feedback. We have also uploaded our demonstration for your review. Please feel free to reach out if you have any questions or require further clarification.

---

### Author Rebuttal · Authors · 2023-08-09

# Section 0 - Author rebuttal to ACs

Dear Reviewers, we truly appreciate your thoughtful review, which has been immensely valuable in refining our paper. Your insights have contributed significantly to the enhancement of our work.

## 0.1 Experiments on HyperNeRF
In order to comprehensively evaluate the performance of our method, we undertook an additional experiment using the HyperDataset. This dataset offers several hundred frames for each scenario, enabling a thorough assessment. Our findings from this experiment reinforce that our algorithm consistently exhibits improvements across nearly all scenarios. Some of the experiment's results are in the reviewer J3AP (Sec. 1.1.2) and you can find the full results in the PDF attached.

## 0.2 Video demo

Following NeurIPS 2023's guidelines for responding, we're able to provide the video by sharing a link in the official comment to the Associate Chair (AC). As a result, we've included the link to our demonstration video in our "official comments" to ACs. This video showcases our perspective on the HyperNeRF dataset. In the video, we start by training our model and NSFF (The second best-performing model on the HyperNeRF dataset) using the left video. Then, we show how NSFF generates the middle video by keeping the view constant but changing the input time. Similarly, the right video is generated by DynPoint, where the view is fixed, but the input time changes.

## 0.3 Qualitative results on Nerfies and iPhone

To better illustrate our algorithm, we've included additional examples of the results it produces on the Nerfies and iPhone datasets in our attached PDF (Figure 1 for Nerfies and Figure 2 for iPhone). When comparing DynPoint and NSFF on the Nerfies dataset, we can clearly see the noticeable differences, which align with the findings we presented in our main paper. Moreover, the comparison on the iPhone dataset showcases our algorithm's edge in handling lengthy and intricate dynamic scenarios. Moreover, we recognize that the iPhone dataset poses a unique challenge due to its intentional design. This design intentionally limits the available multi-view information for each frame. As a result, this complexity makes it more challenging for both the correspondence learning and 3D reconstruction processes.

Once again, we extend our sincere gratitude for your insightful review, and we are fully committed to addressing any further inquiries or suggestions you may have.

---

### Author Response · Authors · 2023-08-14
**Demo video to all reviewers**

Dear ACs & Reviewers,

To address the reviewers' queries, we've prepared a demonstration video showcasing our algorithm's performance on the HyperNeRF dataset. You can access the video through this anonymous link: https://drive.google.com/file/d/1g7AdPvE6SKzZvWnuKfv6hUNW8z0freHE/view?usp=sharing

Best regards,

Authors of Paper 724

---

### Author Response · Authors · 2023-08-20
**Demo video 2 to all reviewers**

Dear Reviewers and ACs,

To enhance the validation of our model's performance, we conducted three distinct experiments. Firstly, we generated a "novel-view" video, maintaining the initial frame's motion while rendering around the camera pose. Secondly, a "stabilized-view" video was produced by keeping the camera pose of the first frame consistent throughout the entire replay. Lastly, we crafted a "bullet-time" video by maneuvering the camera around the pose of the initial frame during playback.

The "novel-view" experiment was carried out on both the curls and tail scenarios of the Nerfies dataset, comparing NSFF and DynPoint. Notably, NSFF demonstrated the second-best performance based on Tab. 2 (Main text).

In the case of the "stabilized-view" experiment, the scenarios 3d-printer and chicken (link in our last comment) from the Hypernerf dataset were utilized. Here, we also compared NSFF and DynPoint, with NSFF showcasing the second-best performance according to Tab. 1 (Rebuttal PDF).

Lastly, the "bullet-time" experiment was executed on the iPhone dataset, particularly on the pillow and mochi-high-five scenarios. A comparison between Nerfies and DynPoint was made, with Nerfies achieving the second-best performance per Tab. 3 (Main text).

Link: https://drive.google.com/file/d/131bIp_wOM8H9Nsqv56L-0VNdn6vNKfJd/view

Best,

Authors of Paper 724

---

### Decision · Program_Chairs · 2023-09-21

**Decision:**

Accept (poster)

**Comment:**

This paper presents a DynPoint for novel view synthesis for Monocular video. This paper is rated as 6/3/7/7. Most reviewers acknowledged the technical contributions of the paper. In the rebuttal period, the authors and the reviewers have in-depth discussions. Most of the reviewer's questions have been addressed. The remaining concern of the reviewer who rated this paper as 3 is about the insufficient demos, and the authors have promised to do more video comparisons and release the videos later.  Thus I recommend accepting the paper. But I also strongly suggest authors show more video results of different methods in the supplementary.